EQUITY, DIVERSITY AND INCLUSION

# Alternative strategies for closing the award gap between white and minority ethnic students

**Abstract** In the United Kingdom, undergraduate students from Black, Asian and other minority ethnic backgrounds receive lower classes of degrees than white students, and similar gaps have been reported in other countries. Such award gaps limit ethnic diversity in academic research, and in the broader job market. Whilst there has been some success in reducing the award gap in the UK, a significant gap still persists. Here, based on an analysis of students studying cell biology at University College London, I show that differences in performance at exams contribute significantly more to the award gap than differences in performance in coursework. From these data, plus scrutiny of the literature, alternative policies are identified to speed up the closure of the award gap and to remove the barriers that prevent students from Black, Asian and other minority ethnic backgrounds from progressing to PhD and postdoctoral positions.

**LOUISE CRAMER\***

**\*For correspondence:**
l.cramer@ucl.ac.uk

**Competing interests:** The author declares that no competing interests exist.

## Introduction

In the United Kingdom (UK), 81.4% of white students and 68.0% of Black, Asian, and other minority ethnic (BAME) students are awarded good undergraduate degrees (where good means a first or upper second), which equates to an award gap of 13.4% (see Methods for more details on the use of the term BAME in this article). The existence of an award gap has been known about for over 25 years, and such gaps have been reported for every university in the UK (*AdvanceHE, 2020b*; *Woolf et al., 2011*).

Similar gaps have also been reported in the Netherlands, United States (US) and Australia for undergraduate and postgraduate taught degrees, and other college-level courses (*Stegers-Jager et al., 2016*; *Kleshinski et al., 2009*; *Tekian et al., 2001*; *Alexander et al., 2009*; *Harris et al., 2020*; *Haak et al., 2011*; *Rath et al., 2012*; *Farmer, 2017*; *Kay-Lambkin et al., 2002*). In theory, scores in qualifying entrance exams may explain the gap in grades awarded. Indeed, studies in the US, UK

and the Netherlands have reported gaps in the marks awarded to different ethnic groups for various qualifying exams for both undergraduate and postgraduate courses (*Miller and Stassun, 2014*; *Miller et al., 2019*; *Reeves and Halikias, 2017*; *Stegers-Jager et al., 2016*; *UK Government, 2020*). However, when tested directly in both the UK and the US, the award gap between ethnic groups persists when comparing students entering higher education with the same qualifications (*Amos and Doku, 2019*; *Tekian et al., 2001*). Furthermore, despite students from Chinese and Indian ethnic backgrounds outperforming white students in university entrance exams in the UK, an award gap of 5% still exists for both of these minority groups (*UK Government, 2020*; *AdvanceHE, 2020b*).

These analyses indicate that institutions themselves are responsible for the award gap between white and BAME students. Within the context of marking students' work, racism or

unconscious (implicit) bias is unlikely since most marking is blind, and when it is not (such as in clinical work) it has been found to be an unlikely major cause (*McManus et al., 2013*; *Woolf et al., 2011*; *Yeates et al., 2017*). Other studies investigating the cause of the award gap have controlled for over 20 other potential factors, such as poverty, age, prior school type and learning style. Yet, while some of these factors have been found to make a contribution, a large, unexplained award gap still remains (*Broeke and Nicholls, 2007*; *Woolf et al., 2013*).

The award gap follows BAME students into their careers: the proportion of white graduates in employment or further education one year after graduation is a third higher than the proportion of BAME graduates (*UK Government, 2020*). Moreover, only 9.1% of professors are BAME (*AdvanceHE, 2020a*), compared with 14% of the population in England and Wales, and 40% in London (*White, 2012*). Most of this attrition takes place at the transition from undergraduate to PhD student, and the transition from PhD student to postdoctoral researcher (see Methods for the trajectory; *AdvanceHE, 2020a*; *AdvanceHE, 2020b*). In practice, selection for interview for the most competitive and prestigious PhD positions – which appear to increase the chances of progressing further in an academic research career – tend to go to those students with the highest grade (that is, to students with a first, or students with an upper second overall with a first or high upper second in the relevant modules), which fewer BAME students possess (*AdvanceHE, 2020b*; this report). This in turn affects progression into postdoctoral positions as students on prestigious PhD programs have a higher chance of obtaining the skills and outputs favoured by selection panels.

There is little direct data, but this notion is consistent with the observation that Black Caribbean PhD students are less likely than other PhD students to have a fully-funded UK Research Council Studentship (a pre-doctoral fellowship that is considered to be highly competitive; *Williams et al., 2019*). It is important to note, however, that other inequalities linked to the undergraduate degree, such as disparities in information on application processes and prior university attended, also likely contribute to reducing the chance of a BAME student progressing to PhD (*Williams et al., 2019*) or receiving an interview for doctorate positions. Furthermore, previous data have shown that other inequalities also contribute to career outcomes in the UK: for example, considerably fewer school leavers (of all white and minority ethnic groups combined) who live in the most deprived neighbourhoods progress to higher education (*Sosu and Ellis, 2014*; *UCAS, 2019*), and Pakistani, Black, or Roma/Gypsy/Irish/other traveller ethnic groups are awarded fewer high grades in end-of-school exams (*UK Government, 2020*).

A similar effect is also seen in the US, where the ethnicity award gap (and to a lesser extent, gender gaps) in the graduate record exams (GRE) leads to higher rejection rates for these minority groups when applying to PhD positions, or even prevents them from applying (*Hodapp and Woodle, 2017*; *Miller and Stassun, 2014*). Efforts to reduce racism and unconscious bias are critical for all stages of recruitment and promotion in academia. However, the transitions to PhD and postdoc positions show by far the largest loss of talent of minority ethnic groups, so focusing resources on these transitions is likely to have the largest impact.

One strategy for reducing the unexplained award gap for undergraduate degrees is to seek quantitative (see Methods for definition) and scientific data on the underlying causes. Although such evidence is recognised as important, it is not widely used (*Amos and Doku, 2019*). Furthermore, the fact the gap can vary from subject to subject – from 3% to 22% in the UK (*AdvanceHE, 2020b*) – is often not considered. This suggests that different solutions (or different combinations of solutions) may be needed for different subjects.

Here, data from courses in cell biology at University College London (UCL) are used to explore the origins of the unexplained award gap, and to suggest measures to reduce this difference and lessen the impact of this gap on academic career progression. In addition, conclusions based on discussions with the organisers of PhD programmes are presented in the article along with an analysis of published initiatives that have increased the recruitment and retention of doctoral students from minority ethnic backgrounds.

## An exploration of the award gap in cell biology courses at University College London

Undergraduate degree pathways in the biosciences and natural sciences at UCL are complex with many possible routes, which is typical across the life sciences in the UK. The cell biology courses (also termed modules) in this study were part of 17 different degree pathways: 15 bioscience and natural science routes were taken by 96.1% of students and the other two routes were taken by the remaining 3.9% (other life science or medical students). The third and often final year of the degree is fragmented into smaller specialised courses, which means that students do not typically take the same modules across all years of their degree. To accommodate this constraint, two complimentary approaches were taken to study the award gap in cell biology.

The first approach, termed the 'year 1–2 pathway', analyses the marks of students entering UCL between 2013–2016 who studied the same cell biology courses in year 1 and 2 (up to 344 students; *Table 1*). Out of the 344 students, eight did not have a record for having taken year 1 cell biology: based on when they joined UCL, five are likely to have transferred directly into year 2 from another UCL degree, and the other three from another university. In year 1, each student takes one of two courses, and the grades awarded were combined as a single year 1 cell biology output. In year 2 all students study the same course.

The other experimental approach, termed the 'year 3 study', looked at four different year 3 courses (448 students, entering UCL between 2012–2016; *Table 1*). Each course had a different composition of students and was chosen on the basis of their class size and variety of specialisms. From the data, it was possible to approximate how many students from the year 1–2 pathway were present in at least one of the year 3 courses (see Methods). This suggests that ~41% (182/448) of the students in the year 3 study flowed from the year 1–2 pathway and an additional ~59% (266/448) came from other bioscience degrees or another degree route.

In the year 1–2 pathway there are four cohorts (one for each year of entry from 2013 to 2016). For clarity, all years in this study are academic years (which in the UK start in September-October and end May-June the following year) and cohorts are cited by the year they start. As each cohort takes two courses – one in year 1 and one in year 2 – a total of eight sets of marks were analysed (*Table 1*). The year 3 study encompasses students who entered UCL from 2012 to 2016, equalling five-year groups in total. Due to the small class sizes, the marks of the five-year groups included in the study are aggregated together for each course.

In the UK, most undergraduate degrees are either a three year bachelor's degree, or a four year undergraduate master's degree (exceptions include medicine). The fourth year was not studied here, because most of that year is a lab project and the ethnicity of students is visible to the examiners, whereas in the cell biology courses selected, all the marking is done blind or by a machine. This circumvents the possibility of unconscious bias or racism contributing to or explaining any gap identified.

In addition to the 792 students included in the study (across both experimental approaches), an additional 22 students were excluded: 12 withheld their ethnicity and 10 (five BAME and five white) did not complete the course. 47.0% (372/792) of the cell biology students assigned themselves as having BAME ethnicity, and 63.3% (501/792) were UK-domiciled. Out of the 792 students included in the study, 95.5% (756) had a grade returned for all the individual components in their course; all data that was returned for all 792 students were included (see Methods - Student Count for further details). In the UK, the final degree is then classified into bands (grades) according to the percent awarded: first class, 70% or higher; upper second (2i), 60–69%; lower second (2ii), 50–59%; third class (40–49%); lower than 40% is further separated into referral and fail. A mark of 60% or higher (i.e., a first or upper second) is considered a good degree or a good mark when looking at components of a degree.

The average award gap in good grades is the difference in the mean proportion of BAME and white students that are awarded good marks. For each course, the final mark awarded is the sum of a weighted exam and weighted coursework mark (with exams typically receiving the higher weighting). Here, to enable a comparison of exam and coursework results, the raw (unweighted) marks are reported for each, along with the final weighted mark.

### Comparing the award gap of UK domicile and international students at UCL

Statistics agencies in the UK report the ethnicity award gap for undergraduate students that are domiciled in the UK, but universities themselves

**Table 1.** Cell biology cohorts studied in this report.

(A) The year 1–2 pathway includes four cohorts that entered UCL between 2013–2016 and took a year 1 and year 2 course in cell biology. *>99% of students stayed with their cohort as it progressed in the pathway. Results for the other ~1% were allocated to the year they took the course. (B) The year 3 study includes students that entered UCL between 2012–2016 and took one of four courses. For each course, the marks of students from all five years groups are aggregated together. All academic years start in October and end June the following year. Final year is based on three year bachelor's or four year master's undergraduate degree.

**A**

| Year 1–2 pathway * | Year of entry in to university | Start of the academic year in which year 1 course was taken | Start of the academic year in which year 2 course was taken | Start of the academic year in which final year was taken |
| --- | --- | --- | --- | --- |
| Cohort 1 | 2013 | 2013 | 2014 | 2015 or 2016 |
| Cohort 2 | 2014 | 2014 | 2015 | 2016 or 2017 |
| Cohort 3 | 2015 | 2015 | 2016 | 2017 or 2018 |
| Cohort 4 | 2016 | 2016 | 2017 | 2018 or 2019 |

**B**

| Year 3 study | Year of entry in to university | Start of the academic year in which year 3 course was taken | Start of the academic year in which final year was taken |
| --- | --- | --- | --- |
| Course A | 2012–16 | 2014–2018 | 2014–2019 |
| Course B | 2012–16 | 2014–2018 | 2014–2019 |
| Course C | 2012–16 | 2014–2018 | 2014–2019 |
| Course D | 2012–16 | 2014–2018 | 2014–2019 |

can collect data on all their students. At UCL, the gap for its UK students is 4.9% on average based on the marks of 8,044 undergraduate students that started their graduation year between 2016–2018, which are similar cohort years to those in this study (*Table 1*). This is at the smaller end of the scale compared to the average gap for all UK students (13.4%). In these same UCL cohorts, the average gap between BAME and white international undergraduate students (N = 5,671 total) is 6.3%, which is near the 4.9% for UK students. This suggests that if any domicile effect exists, it is small. Studying any potential effect of domicile on the gap is not the purpose of this investigation and has been discussed in part elsewhere (*Stegers-Jager et al., 2016*; *Woolf et al., 2013*). In this study, which is designed to find major contributions to the gap that would benefit all students, both UK domicile and international students are included for all the cell biology courses investigated and comparative UCL-wide data.

### Entrance qualifications for cell biology students in this study

The most common type of entrance exam (termed advanced level, or A-level) was accessible from UCL records. At UCL about 90% of UK and 30–40% of international students are admitted with A-levels. The university places strict equivalency requirements on those students that enter with other types of admission exams. Thus A-level grades are a reasonable approximation of all students in the study.

UCL released the A-level grades of a sub-set of students in this study. However, these are representative, and include the first three cohorts (out of four) from the year 1–2 pathway (250 students), 54% of which (135) also took one of the courses in the year 3 study. 72% (179/250) were admitted with A-levels. Typically, only three A-levels are required in sciences or sciences and maths for entrance in science-related degrees. Approximately a quarter received four or more A-levels, and the other three-quarters had three A-levels. To make the analysis fair, for students with four or more entrance subjects, their three highest grades in most relevant A-level subjects (sciences and maths) were included. A student with grades in only two A-level subjects was also included in the analysis.

A-level grades awarded are alphabetical with A* the highest and E the lowest. Each grade was then assigned a numerical value based on the UK standard tariff: A* = 56; A = 48; B = 40; C = 32 (no student had lower than a B). An average of the best three grades (as described above) was then calculated for each student.

The mean admission tariff for BAME (50.53 ± 0.15) and white (50.29 ± 0.33) students included in this study were virtually identical

(*Figure 1A*): the equivalent of nearly all students in each of these ethnic groupings gaining one A* grade and two A grades (which would be a mean tariff of 50.7). These data are very similar to the mean entrance tariff for all BAME (50.27 ± 0.06) and all white (50.03 ± 0.01) undergraduate students who entered UCL with A-levels for the same cohort years as these cell biology students (2013, 2014, 2015 entrants).

An alternative approach for assessing admission qualifications is to look at the distribution of the total number of each individual grade (A*, A, B, C or lower) awarded to each cohort. As expected distributions for BAME and white students were nearly identical (*Figure 1B*); of the total grades awarded to BAME students 95.45% ± 1.01 were A* or A, and for white students this figure was 95.02% ± 0.62 (*Figure 1C*).

Looking at the grades awarded to the same three cohorts of students in year 2 of their biosciences, natural sciences or other degree revealed that proportionally fewer of these same BAME students were awarded good grades for cell biology (*Figure 1C*). The mean of all marks awarded was a low 2i (60–64%) for BAME students and a high 2i (65–69%) for white students (*Figure 1C*). Students admitted with A-levels performed similarly in cell biology to all students admitted in the study as expected (*Figure 1— source data 1*).

These data indicate that prior qualifications do not explain the subsequent award gap in cell biology, at least at the resolution that the A-level grading system permits. This is in line with other admission studies at UK universities (*Amos and Doku, 2019*; *Broeke and Nicholls, 2007*; *Woolf et al., 2013*).

### *Dissecting cell biology course components at the level of good grades*

In all year 1, year 2 and year 3 courses in the study, proportionally fewer BAME students were awarded final good grades than white students (*Figure 2A, B and E*). That the award gap has appeared by year 1 has also been reported for the University of Nottingham (*Amos and Doku, 2019*) and is the case for all undergraduate degrees at UCL (*Figure 2F*). The average gap in good grades awarded each year of study in cell biology (~8–13%) and across UCL (~7–11%) were similar (*Figure 2*, compare E and F, final mark). This suggests that the gaps identified in cell biology are in the range expected for students studying at UCL. Note the UCL award gap each year of study (starting the year 2016–18) were higher (~7–11%; *Figure 2F*) than the final degree

classification (average 5.7%; 13,715 UK and international students starting the graduation year 2016–2018). One likely reason is that for most degrees, a defined number of worse performing courses are automatically excluded when calculating the final classification.

For 11 out of 12 sets of marks studied, this award gap seemed largely derived from the exam component of the course (*Figure 2C–E*). The average exam gaps were similar in year 1 and 2 at over 13% each, and most pronounced in year 3 at nearly 17% (*Figure 2E*). Whether the peak in year 3 is a trend needs to be more widely tested. No exam gap was detected in year 2, cohort 4 (*Figure 2C*). However, a gap overall was still observed for this group (*Figure 2A*). This was explained by a longer 'tail' of lower exam marks for BAME students which resulted in fewer BAME students (61.3%; 23/36) being able to use their coursework grade to obtain a good final mark compared to white students (88.5%; 23/26).

The coursework award gap in good grades was smaller than the exam gap in all individual cohorts and cell biology courses in year 1 (4/4) and year 3 (4/4); and most of the cohorts in year 2 (3/4) (*Figure 2C,D*, compare solid and dashed lines). The coursework award gap was 7.1- (year 1), 13.1- (year 2) and 3.1- (year 3) times smaller than the exam award gap on average (*Figure 2E*, compare exam and coursework values). Taken together this suggests a reproducible and therefore significant pattern.

It is speculated that the small coursework award gap (1–2%) for the year 1–2 pathway in this study reflects small fluctuations in different admission years: this is consistent with 5/8 incidences where BAME students were awarded equal, near equal or slightly higher proportion of good marks for the coursework than white students (observable in *Figure 2C*, compare dashed lines). In contrast, although different admission years cannot be evaluated in the year three study as they were necessarily aggregated (see An exploration of the award gap in cell biology courses), there is no such fluctuation, suggesting that the coursework effect in year 3 courses in this study (average 5% award gap) is consistently small (*Figure 2D*, compare dashed lines).

In theory, if most students are awarded a good grade that may mask a gap. However, when the grades awarded for the exam and coursework were similarly high as each other (such as cohorts 1 and 3 in year 1 for white students) a relatively large award gap between

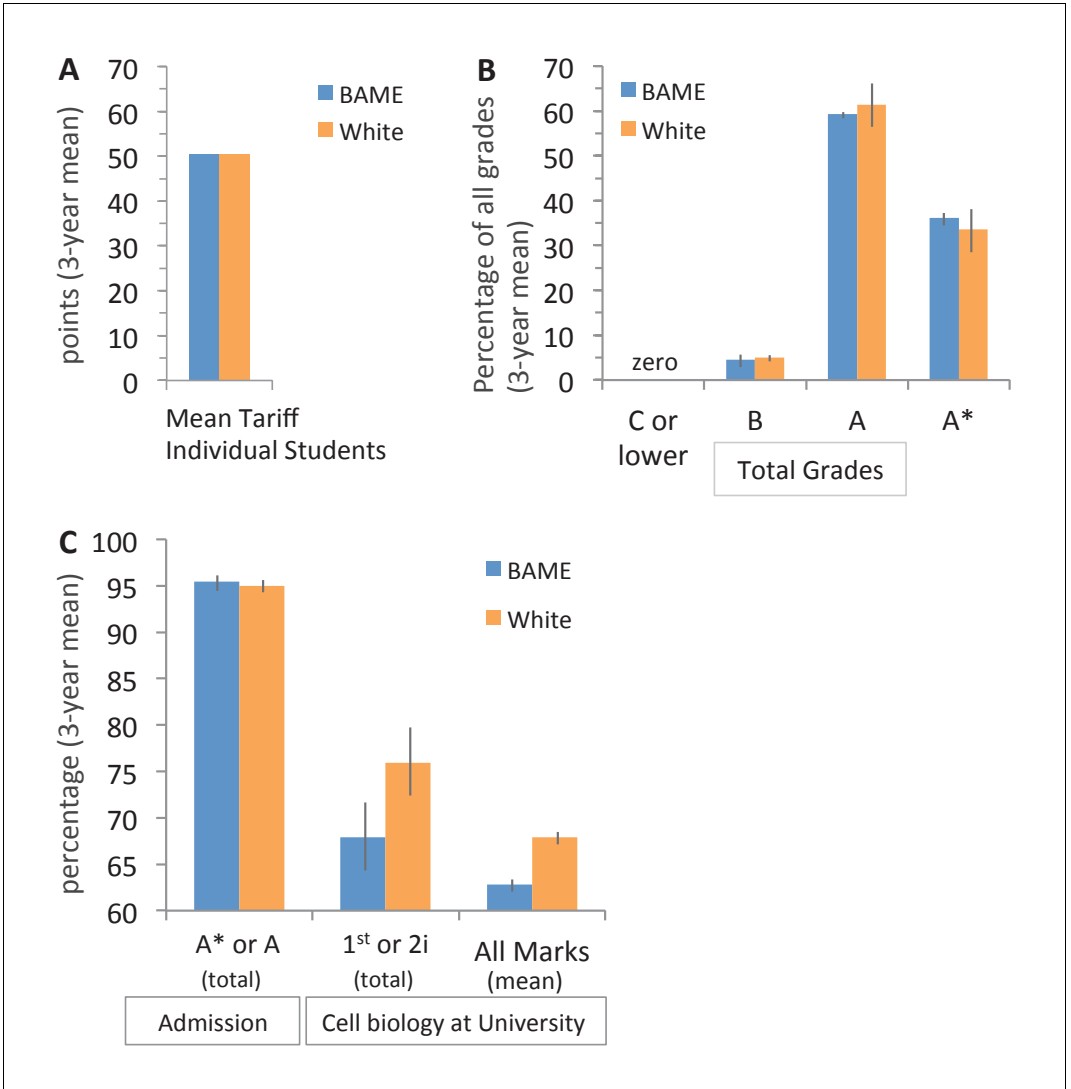

**Figure 1.** Entrance qualifications are near identical for BAME and white students in this study. (**A**) Top three A-level grades awarded to the first three cohorts in year 1–2 pathway that entered UCL between 2013–2015: grades were converted to the UK standard points tariff (see Entrance qualifications for cell biology students in this study), the average for each student was calculated, and the mean of all three cohorts was determined (SEM: ±0.15 for BAME and ±0.33 for white students, which are too small to see on the graph). N = 179 students (82 BAME, 97 white); 536 entrance grades (246 BAME, 290 white). (**B**) Percentage of grades awarded to students in (**A**) that were C or lower, B, A, or A*: 3 year mean (± SEM). (**C**) The percentage of top admission (A* and A) grades awarded to BAME and white students in (**A**), the percentage of students that received the top two grade bands (first or 2i) in year 2 of cell biology in 2014–2016 (same students shown in (**A**)), and their average mark in year 2 of cell biology: each column: (3 year mean ± SEM).

The online version of this article includes the following source data for figure 1:

**Source data 1.** Source data on the entrance qualifications of white and BAME cell biology students entering UCL in 2013–2015 and their subsequent grades for year 2 cell biology.

white and BAME students was still distinguishable for the exam, but was small or none for the coursework of the same cohort (*Figure 2C*). Similarly, it is unlikely that larger gaps are only revealed when tasks are harder: for example, in year 3, which is expected to have harder coursework tasks overall, grades were broadly the same for exams and coursework (for white students), yet the gap in grades awarded to BAME and white students were only evident in the exam.

Overall, the method of looking at good marks seems sensitive enough to both detect a gap and to reflect its size if it exists. Cohen's D

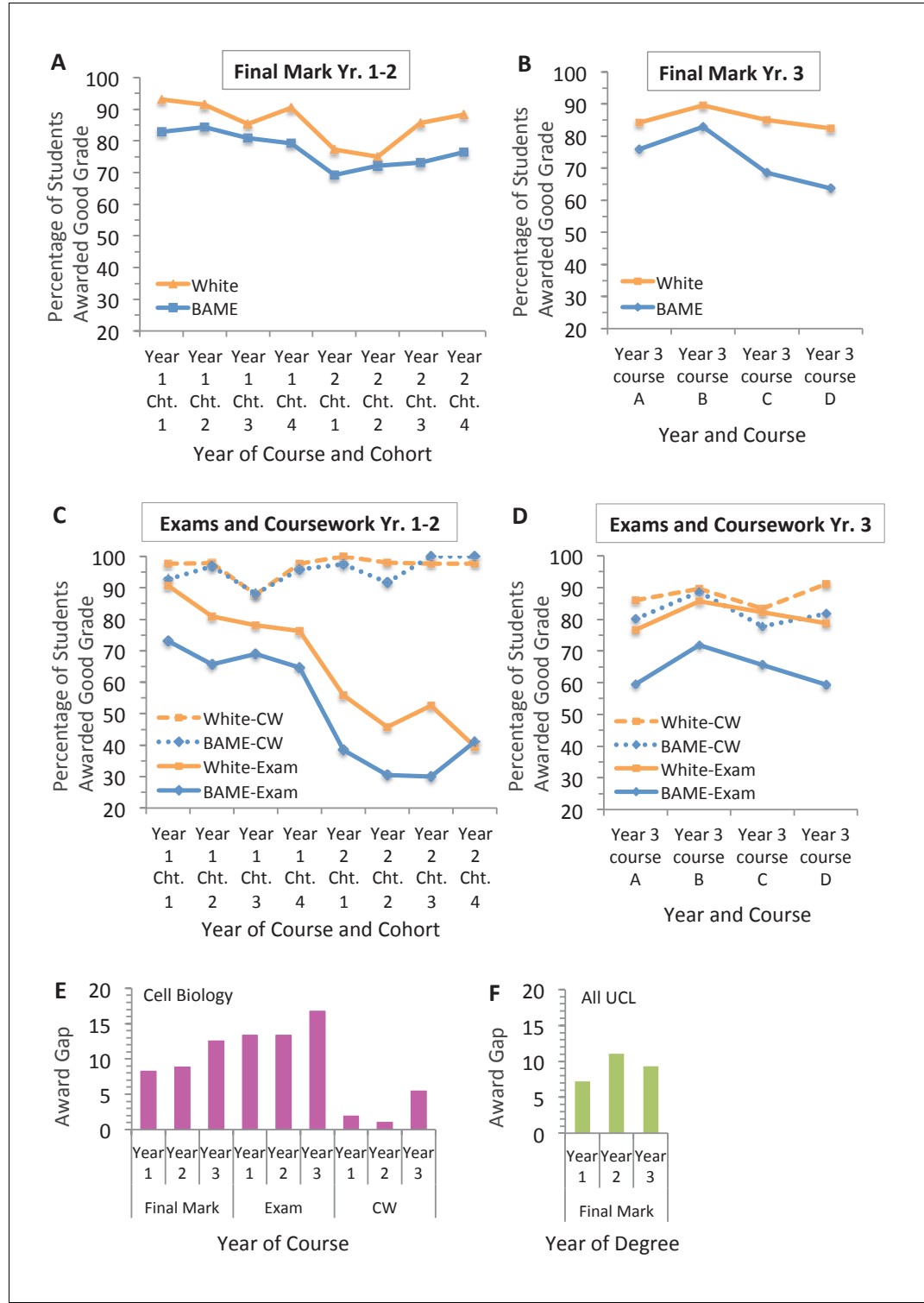

**Figure 2.** The exam is the main contributor to the award gap in cell biology. Percentage of BAME and white students in each cohort (shown in *Table 1*) that were awarded good grades in cell biology for the final course mark in the (**A**) year 1–2 pathway and (**B**) year 3 study; and the exam and coursework components in the (**C**) year 1–2 pathway and in the (**D**) year 3 study. (**E**) Difference in the percentage of white and BAME students awarded a good mark (first or 2i) for each year of the cell biology course (data shown in A-D) based on the mean of all four cohorts or courses (means and SEMs are reported in *Figure 2—source data 1*). (**F**) Difference in mean proportion of good grades awarded to all white and BAME students at UCL in 2016, 2017 and 2018 for each year of study

*Figure 2 continued on next page*

*Figure 2 continued*

(years 1, 2 and 3). Total number of students that completed the course = 792; in the year 1–2 pathway = 344 (167 BAME, 177 white); in the year 3 study = 448 (205 BAME, 243 white). 22 students were excluded (12 due to unknown ethnicity; 10 did not complete the course).
The online version of this article includes the following source data for figure 2:

**Source data 1.** Source data for the final course, exam and coursework marks of white and BAME students in the year 1–2 pathway and year three study; and the grades for all UCL undergraduates that took year 1, 2 and 3 in either 2016, 2017 or 2018.

(which is a measure of effect size) was then applied to the mean award gap of all four cohorts or courses studied for each component in the year 1–2 pathway and year 3 study. This revealed that the effect for coursework is three to six times smaller than the exam (*Figure 2—source data 1*).

### Testing for potential hidden award gaps across all marks

There is a further possibility that there may be a larger coursework award gap, but it was not detectable because it occurred within a specific range of marks. However, in year 2 and year 3 cell biology a larger coursework gap could not be found when comparing every absolute mark awarded to BAME and white students at the resolution of 5% intervals (which is one-half grade band) from less than 30% to 100% scores (*Figure 3*; only cohorts 1–3 were included in the analysis for year 2 – see legend for more details) Clearly evident in these frequency distributions, the peak of final marks (*Figure 3A and B*) and the peak of exam marks (*Figure 3C and D*) awarded to BAME students were shifted half to more than one grade band lower than white students (*Figure 3A–D*). The end result is that BAME students are most likely to receive a low 2i (60–64%), whilst white students are more likely to receive a high 2i (65–69%) or a low first (70–74%) for these courses overall (*Figure 3A and B*).

In contrast, the histograms of coursework marks for white and BAME students were almost completely superimposed in both studies (*Figure 3E and F*). Further inspection of the curves demonstrates the method is sensitive enough to visualise the small award gap in good marks between BAME and white students for the coursework in year 3 (*Figure 3F*, slight broadening in the bell-shaped, blue curve away from the orange curve at 59% to 40%) that was reported in *Figure 2E*. However, clearly there is no larger visible gap anywhere in the entire spectrum of marks at the resolution of 5%

intervals for either year 2 or 3 cell biology coursework (*Figure 3E and F*).

There was insufficient data to fully assess year 1 cell biology course components by first class marks, though they were found to follow the same general trends as year 2 and 3 (see Limitations of the study).

### Cell biology award gaps in first class marks

As expected from these histograms (*Figure 3*) the average award gaps for year 2 and year 3 coursework at the level of first class marks-only remained small (2.60% and 4.11%, respectively) similar to that reported at the level of all good marks for these modules (2.14% and 5.45%; *Figure 3—figure supplement 1*, compare A and B).

The average exam award gaps for first class marks in year 2 and 3 (5.60% and 9.29%) were smaller than those for good marks (18.37% and 16.75%), but importantly were two (or more) times larger than the coursework gaps at the same level (*Figure 3—figure supplement 1*, compare A and B). This pattern of the exam award gap in first class marks being larger than the coursework award gap was also observed in three out of four of the year 3 courses (B, C and D); in course A, both the exam and coursework award gap were similarly small (*Figure 3—figure supplement 1C*). There were too few white (and BAME) students awarded first class marks for the exam to compare individual year 2 cohorts (observable in *Figure 3*; see *Figure 3—figure supplement 1—source data 1* for comparison of year 2 course components, aggregating the cohorts and Methods for more details).

In addition, a number of factors contribute to the relative size of the award gap for the final cell biology mark: the weighting for the coursework and to what extent the coursework mark can lift a student to a final grade band that is higher than that for the exam mark, and how far the exam is shifted to lower marks. All these factors can vary by course (some are observable for example in *Figure 3*); and in these data the gap in first class marks awarded for the final module mark was greater than all good marks in year 2,

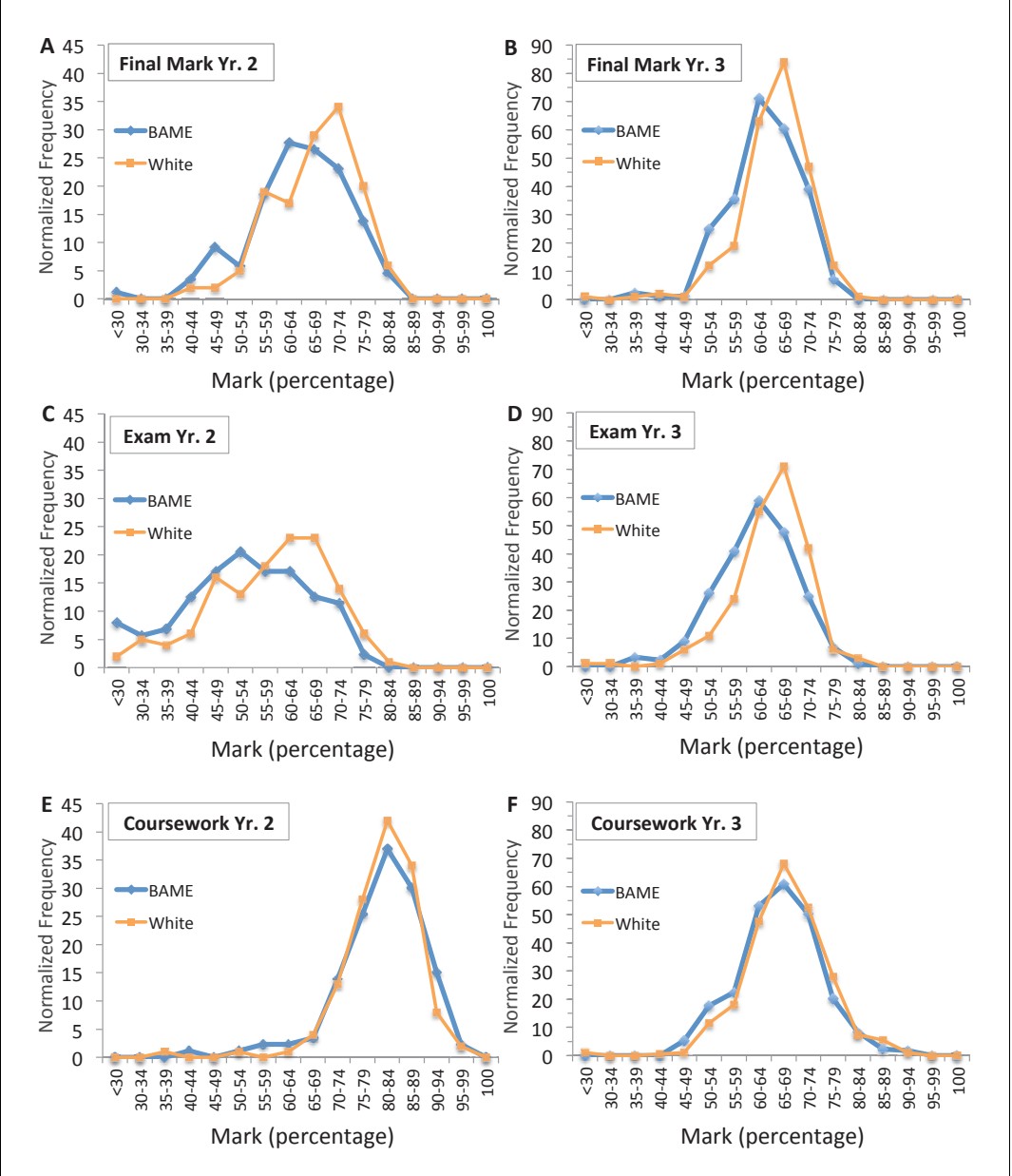

**Figure 3.** Comparison of all marks does not reveal any larger award gap in the coursework. Comparisons of all marks awarded to individual BAME and white students in year 2 (A, C, E) and year 3 (B, D, F) cell biology for all courses in year 3 study and three cohorts (cohorts 1-3) in year 2; cohort 4 in year two was excluded since there was no exam award gap in good marks, as reported (*Figure 2C*; and main text) and also none in first class marks-only. For each component, frequency of marks for each interval was aggregated across cohorts. To enable fair comparison BAME, student frequencies were then normalised to the total number of white students in that component (normalisation factors are shown in *Figure 3—figure supplement 1—source data 1*). Total number of students that completed the course = 698; in year 2 = 250 max (116 BAME, 134 white); in year 3 = 448 max (205 BAME, 243 white). Exact cohorts and N for each component in (A–F) is shown in *Figure 3—figure supplement 1—source data 1*.

The online version of this article includes the following source data and figure supplement(s) for figure 3:

**Figure supplement 1.** The award gap for first class marks is similar to the difference in good marks awarded for coursework in cell biology.

**Figure supplement 1—source data 1.** Source data for the values used to calculate the final mark, exam and coursework award gap between white and BAME students in years 2 and 3.

but smaller in year 3 (*Figure 3—figure supplement 1* – compare A and B and quantified in the source data).

These differences are not unexpected. Indeed, although the exam award gap for first class marks being smaller than all good marks is similar to the UK-wide pattern for the final degree, it is evident from other national statistics, and at UCL, that the amount first class and upper second marks contribute to the award gap varies at least between different subjects and minority ethnic groups (*AdvanceHE, 2020b*). It is therefore plausible that different studies will detect different patterns in this regard.

Taken together, these data indicate that in these cell biology courses, the exam is the main contributor to the gap in marks awarded for the course overall. If there is a contribution from coursework, it is relatively small in all years and may not be consistently present in years 1 and 2.

### Evaluating potential underlying causes of the award gap in exams

There are fewer cell biology courses in the year 1–2 pathway than in the year 3 study, and it is possible that comparing more biology courses in these first two years could identify a larger award gap in coursework, similar to the one found in year 3. On the corollary, the inclusion of students from multiple years groups in the year 3 study might drive the small coursework award gap detected. Future studies will be needed to distinguish between these two possibilities. Furthermore, it is also not possible to determine whether the difference in size of the exam and coursework award gaps between year 1, 2 and 3, and between different year 3 modules, is due to how each course is taught: for example, how the course is delivered, the precise types of assessments, the level of support students receive on different degree routes or a combination of these factors.

Similarly, why exams overall produce a bigger gap than coursework in the year 1, 2 and 3 courses studied is not yet known. The finding is consistent with medicine where different types of assessments produce gaps of different sizes; although, unlike here, exams versus coursework were not explicitly tested (*Woolf et al., 2011*). In cell biology, the exam assessment method in year 1 (mainly multiple choice) is different to years 2 and 3 (mainly essays). One interpretation of this difference is that the process of taking an exam may contribute to the gap. To test how widespread these findings are, other cell biology courses in other universities need to be evaluated.

In the future, it will also be important to directly test the extent to which the examination process, its content or other potential factors matter. This will reveal further precision on the underlying causes of the exam award gap. Related to this, and equally important in its own right, will be to determine if other types of degree subjects have similar or distinct patterns in their own gaps to that observed for cell biology. For example, how does the size and pattern of the award gap in subjects that do not examine by essays compare to those that mainly rely on coursework? Furthermore, the different assessment methods forced by the COVID-19 pandemic may also help identify 'what matters'.

Overall, these results suggest that this strategy of dissecting individual components of a degree could reveal what is causing the award gap for a particular subject. Although a description goes beyond the scope of this paper, these results could inform the development of a new educational framework aimed at reducing the award gap in cell biology.

### Limitations of the study

The study is restricted to certain courses at one university. The wider applicability of the written exam and coursework findings require testing in cell biology in other universities and in other subjects across institutions. Most degree subjects (if not all) are composed of different types of activities and assessed components. The method described here is envisaged to be readily adaptable to look systematically at the relevant components of any particular subject.

In year 1 cell biology there was more variability between cohorts in the proportion of students awarded first class marks for the exam and for the coursework, for both BAME and white students. It is speculated that this was due to the combination of disaggregating good marks further, and the necessity of combining two different modules for each cohort in year 1 which may have varied marking schemes (whereas all cohorts in year 2 and year 3 were composed of a single course). Additional year 1 cohorts are required to test the preliminary finding that the average coursework award gap in first class marks was 1.6-times smaller than the exam gap (for cohorts one, two and three), which is similar to that for the year 2 and 3 cell biology courses, which are 2.2 and 2.3-times smaller, respectively (*Figure 3—figure supplement 1*). In cohort four (year 1), there was no

award gap in first class marks for the exam, which is consistent with the similar finding for the same students in year 2 (cohort four, year 2) at the level of all good marks reported (*Figure 2C*).

Except for this year 1 limitation, the studies here are of sufficient size to assess gaps at the level of the components of an individual course, a main aim of this investigation. However, disaggregating exam and coursework marks of individual minority ethnic groups will require further investigation with larger cohort sizes. In particular, it would be informative to look at students with Black ethnic backgrounds because proportionally these students are awarded the fewest good degrees in the UK (*AdvanceHE, 2020b*). Only 2% of the cell biology students in this study self-identified as Black African, Black Caribbean or other Black background (UK and international domiciled) which is a similar proportion across all undergraduates at UCL (3%, UK and international) for similar cohort years (2012–2016 entrants), and similar to that for England and Wales (3%) at the last census (2011), but underrepresented for London (13%). Preliminary data, which is consistent with the UK pattern, looking at year 3 cell biology courses, the proportion of Black students awarded good grades for the final course mark (~38%) and for the exam (~38%) was up to 2-times lower than all minority ethnic students (BAME) for these two components (72.7% ± 3.7% and 64.1% ± 2.6, respectively). However, the proportion of Black students awarded good grades for cell biology coursework (~87%) was similar to all BAME (82.1% ± 2.0) and white (87.5% ± 1.5) students.

## Policies for reducing the award gap in undergraduate degrees

For the past 10 years, only 0.5% per year has been shaved off the undergraduate degree award gap (*AdvanceHE, 2020b*). Yet in the same period there have been extensive reviews and reports that have detailed why the unexplained, award gap exists and hundreds of recommendations on how universities can tackle it (*Alexander and Arday, 2015*; *Amos and Doku, 2019*; *Berry and Loke, 2011*; *ECU and HEA, 2008*; *Miller, 2016*; *NUS, 2011*; summarised in *Figure 4*; for a review of earlier studies (from pre-1990s) see *Singh, 2011*).

A scrutiny of the literature suggests that various factors are reducing the speed at which the award gap is closing: lack of staff time coupled to insufficient funding and resources to implement existing recommendations (see Policy 1); limited scientific approaches and diverse types of evidence on the underlying causes (Policy 2); limited funding for fundamental scientific studies and for applied scientific approaches on what works (Policies 3 and 4). That the gap is only disappearing very slowly also propels the need for alternative, parallel action to reduce impact on academic (and other) career progression, by removing (in the case of academic careers) barriers to accessing PhD and postdoc positions (Policy 5). Overall, combining these five strategies may help eliminate the award and progression gap between white and BAME students.

### *Policy 1. Create time and fund resources to develop, implement and monitor impact of recommendations*

Many universities, including UCL, have put considerable effort into identifying what resources are needed to close the award gap, with nearly 200 recommendations in three reports alone (*Figure 4*; *Box 1*). But therein lay two fundamental issues: they are all hugely time-intensive to develop and implement, both for academic and other university staff, and cost-heavy for universities. Academics already work excessive hours just to get their 'regular' job done (*Richardson, 2019*; *UCU, 2016*), yet seem directly responsible for delivering many of the recommendations. The UK does not appear in the world's top 100 for staff-student ratios (*THE, 2020*) and the best UK ones are misleading, as they do not measure proportion of time spent on teaching or with students. Even if it is still 'too early to tell', the same work-load issues and ownership of who develops and delivers the actions apply. The complex mix of recommendations distils to a simpler accelerated answer: increase staff ratios and remove other speed barriers.

(1a) Increase the number of academics per student and monitor the new ratio to keep pace with the rise in student numbers
This would enable identification of sufficient academic staff that can be specifically tasked with a remit to work on the award gap so more time can be devoted to implementing initiatives aimed at decreasing the award gap. These additional academics should work with module or programme organisers, or both, to develop and implement existing recommendations (*Figure 4*) or design new education frameworks

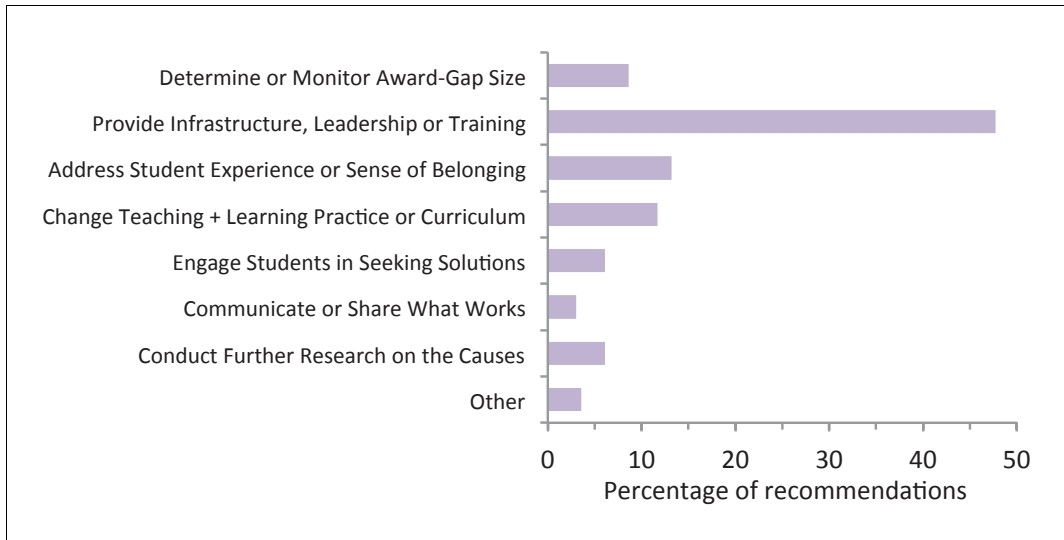

**Figure 4.** Summary of previous recommendations on how to close the award gap between white and BAME undergraduate students. Recommendations are from three major UK original research reports: *ECU and HEA, 2008*; *Berry and Loke, 2011*; *Amos and Doku, 2019*. Reports were selected based on the depth and breadth of the data, the number of participating higher education institutions (HEIs) and representation across the past decade. The recommendations from each report were then assigned to seven different categories based on the language used to describe them. For example, characteristics of 'HEI culture', 'student experience' and 'racism' were grouped together, and 'change curriculum' encompasses recommendations with terms such as diversify, decolonise, internationalise, and make more inclusive. N = 197 recommendations; 168 participating HEIs total: 99 HEIs partnered with the UK National Union of Students and Universities UK; and 69 HEIs worked with the Equality Challenge Unit and the Higher Education Academy (now AdvanceHE).

The online version of this article includes the following source data for figure 4:

**Source data 1.** Source data for the values used to calculate the percentages of the different recommendation categories.

---

incorporating new evidence, such as the findings of this study, that are best suited for each subject.

(1b) Increase the number of staff to extract and analyse data and place more of them locally within departments to work on the gap

This is essential to cope with the time-heavy (but critical) activity of monitoring and measuring impact of local actions. Currently individuals and departments working on the undergraduate award gap must typically rely on a single, central student record team, or similar, to extract student results. Such teams are often small and understandably must instead prioritise their own important tasks, which include complying with statutory regulations and maintaining the student record for the entire university. Furthermore, departmental teaching teams typically do not have dedicated analysts.

(1c) Release ethnicity of students in selected, pre-approved undergraduate class lists

Another speed barrier is that in some higher education institutes – but not all – there is no mechanism for course-organisers (and entire departments) to access ethnicity (or gender) for their undergraduate class lists whilst they are live. Instead, organisers must wait to receive ethnicity data matched to student results retrospectively, which may well not be until the next academic year, delaying results of impact of any actions (see Policy 1b). Students in the UK give permission for their special category data (such as ethnicity) to be used for the legal purpose of monitoring equality of opportunity or treatment. This suggests that collecting this data in real-time will not breach UK data protection and recording of processing regulations.

(1d) Increase funding and resources to implement existing recommendations

Investment is required to enable the increase in staff time or number, or both, and to provide

## Box 1. Examples of specific recommendations provided by previous policy reports on how to close the award gap.

A non-exhaustive selection of some of the types of previously suggested actions within the three largest categories. (1) **Provide infrastructure leadership and training by**: (a) ensuring clear messages on the importance of addressing the problem comes from all tiers of leadership; (b) creating and enabling access to resources for each category of recommendation; (c) providing access to data and reporting transparently; (d) offering staff training on: how to recognise and respond to racism and micro-aggressions on campus, unconscious (implicit) bias training, how to build supportive and inclusive environments, and how to assess and mark fairly. (2) **Changing teaching and learning practices or curriculums by:** (a) diversifying how a subject is taught and assessed through promoting inclusive learning environments, increasing feedback to students, encouraging students to directly engage with their peers and academic staff, providing equal access to feedback and learning opportunities, providing a range of assessments, marking students' work anonymously where feasible and according to specific mark schemes; (b) Identifying, in all subjects, a fair representation of previous contributions to that subject's knowledge-base from different ethnic (and gender) backgrounds, and for relevant subjects or topics create curricula that are more representative of different ethnic (and gender) cultural, historical and ideological backgrounds and experiences. (3) **Addressing students' experience or sense of belonging by** (a) providing diverse spaces and opportunities on campus where communities can thrive in supportive and inclusive environments; (b) celebrating and recognising the importance of diversity; (c) engaging with students equally; (d) equalising differences in cultural capital; (e) addressing racism; (f) providing role models where possible.

resources to implement the many recommendations (*Figure 4*); there are >5 types of training alone suggested for example (*Box 1*). Some universities (including University of the Arts London (UAL) and University of Brighton) have responded by releasing university access and other funds to hire new, permanent staff to work on the award gap, whilst others (including UCL) are enabling temporary 'buy-out' of existing staff time to develop and implement initiatives. This is welcome news; however, the total amounts to a handful of full-time equivalent posts, whereas at a single large university alone there are >4,400 courses (modules) within >440 separate undergraduate degrees (estimate based on the UCL record) and scaling that up to >50,000 undergraduate degrees across the UK (http://www.ucas.com) approaches half a million courses.

The UK government who have tasked universities with removing the difference in degree scores could fund substantially more academic positions specifically for this role, so there is one or more in each department. These positions could be made permanent which would substantially increase the £8 per BAME student provided for training and other resources to reduce the gap (see Policy 3). A proposed stakeholder alliance could also make a significant investment (Policy 4).

### Policy 2. Increase scientific evidence on causes of the unexplained award gap and the diversity of approaches

At first glance it may appear there is plenty of evidence on causes of the award gap, which in turn informs the many suggested actions (*Figure 4*). However, on closer inspection there seem two large gaps in the evidence base. Closing these have the potential to uncover previously unrecognised solutions and to inform which ones work best.

Three comprehensive reports representing the beginning, middle and end of the past decade have comprised nearly 200 recommendations on how to close the unexplained award gap (*Figure 4*). However, only a very small number (estimated 0.51%) of these recommendations appear to be supported by raw numerical data. Clearly much has been quantified, particularly the size of award gaps. However, to date this has almost exclusively been used as evidence that the problem exists, as a benchmark, or as attempts to identify the size of various potential, contributing external factors (see introduction). Yet, these do not identify the larger, university-driven (unexplained) causes of the gap.

The majority of these recommendations appear to be based on qualitative data (estimate 20.3%), such as surveys, interviews and

discussions, particularly those related to the following two categories which represent nearly one-quarter of all the recommendations: student experience of university life and sense of belonging; teaching and learning practice and the degree curriculum (*Figure 4*; see also Methods for other details).

Engaging with the lived experiences and views of minority ethnic students and staff is fundamentally important for developing potential key solutions. In the context of identifying areas that might help speed up gap closure, balancing these with quantitative approaches (see Methods for definition), such as the one in this report, which has identified a possible contributory cause, will increase the diversity of the evidence base.

Robust scientific evidence can be qualitative or quantitative in nature, and both have equal merit and complimentary benefits. A recent comprehensive analysis of what 99 UK higher education institutions are currently doing concluded that 'a far more scientific approach' is one of the top five most outstanding actions required to close the award gap (*Amos and Doku, 2019*). There are enormous benefits of a scientific approach: it clearly identifies whether any particular issue is disproportionate to minority ethnic students (or staff) and is therefore a powerful indicator of possible cause. Yet many (25/30) of the surveys and discussions to date do not seem to report control data (*NUS, 2011*; Methods). Recommendations in the area of tackling racism are not likely much further informed by control groups. However, in the context of speed of implementation for many of the other recommendations (*Figure 4*; *Box 1*), having a more scientific message on possible cause may persuade others to take action, to invest or guide the prioritisation of allocating valuable resources.

Explicitly advocating scientific approaches in the UK's Race Equality charter mark and in Fellow of Higher Education Academy Awards, when either is related to degree award gap work, may also help the rate of progress, as seen with the gender-professor gap and the Athena Swan charter mark.

### Policy 3. Increase direct funding to research teams proposing scientific approaches

Enabling Policy 2 requires investment. Although the UK government has announced universities must remove the gap, it is only providing £3 million/year (government and other sources) in targeted-funding, equivalent to just £8 per BAME undergraduate/year (see Methods). In addition, the commitment is only for 2–3 years; yet the award gap will take several decades to close at current projection (*AdvanceHE, 2020b*). Universities also have separate 'access funds' which could in theory be released to individual research groups. Their general availability, however, seems unclear as well as in competition with other equally important priority areas such as enabling students from disadvantaged backgrounds to attend university.

### Policy 4. Involve different stakeholders to create a funding alliance

Vanishing the award gap should concern everyone. It provides all employers with a major route to considerably close their ethnicity pay- and career-gaps and thus increases productivity through enhanced work-force diversity. This in turn increases government tax revenue through increased corporate profits. Furthermore, gap closure is the sole method available to universities to remove their breach of UK laws on race equality.

A funding alliance seems a pragmatic way to raise and accelerate the large investment sums required for the success all parties stand to gain if the gap is eliminated. The remit could embrace both fundamental scientific approaches (Policies 2 and 3), and applied scientific endeavours to implement and monitor existing recommendations (Policy 1) as well as manage essential funding structures. This works elsewhere; take the Dementia Discovery Fund of £250 million, an alliance of the UK government, global pharmaceutical companies, a UK charity, and global philanthropy which funds research into effective treatments for dementia, equivalent to £294 per UK patient (see Methods).

Strong leaders who believe what society has to gain from such an alliance are required for success. It is proposed then to take advantage of a pre-existing structure such as the UK's Nuffield Foundation, which already seems to demonstrate such a belief both in its strategy document 2017–2021 and in its funding of projects widely across education, welfare and justice; their housing of the suggested alliance may speed up its creation.

### Policy 5. Re-design monitoring and recruitment for early academic career positions whilst the award gap still exists

This policy combines determining exactly where the issue in PhD and postdoc recruitment is for different minority ethnic groups (Policy 5a), implementation of more accurate national data (Policy 5b), and direct action on the impact of an unexplained difference in grades for admission exams and other selection criteria that disadvantage PhD applicants from minority ethnic groups (Policy 5c). Although most of this section focuses on PhD admission the principles of recruitment redesign and access can be applied to postdoc recruitment.

### (5a) Increase the monitoring and reporting of applications, interviews and offers for PhD student and postdoctoral researcher positions by ethnicity and gender

Although at a national level, we know where the ethnicity career gaps are in terms of academic progression and the size of those gaps (described in the introduction), that information alone is not precise enough to design actions with the most impact. For example, preliminary work in our department reveals that Black people are underrepresented at application, whilst the bigger barrier for candidates from Asian ethnic groups seems underrepresentation at interview for both PhD and postdoctoral positions (ethnicity of candidates are not revealed during selection). Where such a monitoring policy is practiced, consequent legal and fair re-design of recruitment criteria combined with other successful initiatives have been powerful (Policy 5c).

### (5b) Report national numbers of PhD students and postdocs separately to other positions

Whilst individual departments have capacity to accurately record the ethnicity of their own PhD and postdocs, local and national data are required to monitor the wider impact of actions. Yet, UK data that are readily available (e.g. from the Higher Education Statistics Agency and Advance HE) do not report ethnicity of PhD students separate from other postgraduate research (PGR) students (such as master of research); nor do they report postdocs separate from other staff levels of similar seniority (see Methods). However, only a PhD typically qualifies as the first academic career step. Using PGR as a benchmark for BAME PhD students and several job groups as a proxy for BAME postdocs

may over-estimate already low numbers. In the US, on the other hand, PhD recipients are separately reported.

### (5c) Widen initiatives that work to increase representation of minority ethnic students on PhD programmes

Some pioneering initiatives in the UK and US have driven a two to four-fold increase in the proportion of PhD students from minority ethnic backgrounds in life sciences and physics over the last 3–6 years (*Hodapp and Woodle, 2017*; Frank Bayliss, Tetrad Programme Office and Nadine Mogford personal communications). Where measured, retention of PhD students is also higher than the national average in these schemes (*Hodapp and Woodle, 2017*). These are the London (UK) Interdisciplinary Doctorate programme (LIDo), Tetrad PhD programme at UCSF, and the American Physical Society (APS) Bridge Programme (which has also increased female doctorates in physics). Proportion of minority ethnic students on PhD programmes overall at UCSF have also increased in the past five years (Office for Institutional Research). There may be other programmes that have also had success unknown to the author. Common to the success of these UK and US programmes is:

i. Removing admission exam criteria known to have an unexplained ethnicity award gap. APS Bridge and Tetrad (and 17/19 of UCSF programmes overall as of August 2020) have removed the Graduate Record Examinations (GRE) test, and LIDo have removed the highest undergraduate degree class (first class) as advantage for selection for interview or award of a PhD place. Many other US PhD programmes, particularly in the life sciences, are also removing the GRE from their selection criteria (*Langin, 2019*; see also https://beyondthegre.org/ for original studies and further information). It is also suggested that candidates' names are removed from applications, which LIDo have had partial success with. However, an issue that remains is how to ensure 100% of referees use applicant number instead of name in their letters of recommendation.

ii. Actively encourage minority ethnic students to apply. Both the LIDo and UCSF schemes have links with undergraduate institutions in which minority ethnic students are more likely to attend (minority-serving institutions in the US and low- and medium-tariff universities in the UK). The APS Bridge scheme links

prospective PhD students with available places at a number of participating universities, supports their applications and mentors them afterwards. An added benefit of these schemes is that they incorporate opportunities for current doctorate students from minority backgrounds to act as role models in such a way that also contributes to their own career development.

iii. Increase the competitiveness of historically marginalised and underrepresented minority ethnic candidates by removing other known barriers to their selection. Both LIDo and UCSF run salaried-summer research training schemes aimed at undergraduate students from minority ethnic backgrounds and other disadvantaged groups. In the UK, proportionally twice as many white (34.9%) than Black (16.3%) students attend the most research-intensive universities, whilst applicants from Asian ethnic groups are slightly more likely to have attended medium-intensive ones (**HESA, 2020**). These figures are almost exactly mirrored in the US: among its American citizens and permanent residents, proportionally twice the number of white (90%) than minority ethnic undergraduates (43%) attend research-intensive universities (estimated from table 2.1 available as separate source data from **NSF, 2019b** and data within **NASEM, 2019**). Yet prior research experience is one of the top criteria for selecting candidates for interview. The APS Bridge Scheme recognises that financial constraints disproportionally lower the competitiveness of applicants from minority ethnic groups and takes that into account when selecting candidates (**Hodapp and Woodle, 2017**).

iv. In part, the success of these three programmes is due to the specific training that their recruitment panels receive, and that some use a common rubric rating scale for selecting applicants. Ethnic diversity on selection panels is also an ongoing concern – it directly relates to the high loss of potential senior academics early in the pipeline to professor (detailed in the introduction), and some universities (including UCL) are implementing initiatives to increase representation of minority ethnic staff on panels.

$10 million from the US National Science Foundation has now been awarded to extend the APS Bridge Programme to chemistry, geosciences, astronomy and material sciences.

In the UK, Research England and Office for Students have also recently announced a joint funding call to support initiatives to increase access and participation for minority ethnic postgraduate research students, initially planned for release in Autumn 2020 (but delayed due to COVID-19).

## Conclusion

Many caring academic and other university staff are driven to eliminate the unexplained, undergraduate degree award gap, but it is an immense challenge. What seems missing is sufficient and sustained investment to create enough time, resources and training for them to do so. In addition, there needs to be more funding (such as through an alliance of stakeholders) for scientific approaches to research the fundamental causes of this gap, and implement and identify which of the existing recommendations work.

Whilst the award gap persists, parallel actions on monitoring and changing recruitment practices are more widely needed across society in all jobs that employ graduates, to accelerate equal opportunity. Combining both sets of actions (Policies 1–5) will widen opportunity for increased numbers of BAME individuals to access the trajectory that leads to the more senior academic positions. This may positively feedback on itself, as students consider the lack of ethnic diversity among senior leaders as the number one cause of the gap (**Amos and Doku, 2019**).

## Methods

### Definition of terms and international variations

In the UK, BAME (and also BME) is a nationally used term to represent all ethnic groups within Black, Asian, mixed ethnicity and other minority ethnic individuals (which includes white minority ethnic groups). It is recognised here, that individuals have different opinions about using these terms. 'Minority' is relative to the country's total demographic. Students select their own ethnicity when they enrol at university, standardised across UK higher education institutions (https://www.hesa.ac.uk/collection/c18051/a/ethnic).

In the US, underrepresented minority (also known as URM) is instead more widely used and can refer to ethnicity or other qualifier; where 'underrepresented' indicates proportionally fewer than the proportion of that group in the

population as a whole. The US has a higher proportion of minority ethnic people (36% of the total population) compared to the UK's 14%. BAME groups are broadly comparable to URM ethnic groups in the US. The exception are Asian groups which, whilst are minority groups in both countries, have variable degree award gaps when compared to white Americans, at least in medicine (discussed in *Kleshinski et al., 2009*), and score higher than white students in some other degree subjects (*Alexander et al., 2009*). Also Asian Americans, like white Americans, are overrepresented in PhD positions (*Hodapp and Woodle, 2017*; *NSF, 2019a*), which is linked to these two groups scoring higher than all other ethnicities (and Asian higher than white) in graduate record exams (*Miller and Stassun, 2014*). Other underrepresented minority ethnic groups also reflect regional populations in different countries.

Here, the term 'award gap', or 'ethnicity award gap' refer to the gap in scores awarded to different ethnic groups for their degree or its components. Note, in some other literature, the term 'attainment' or 'achievement' gap has also been used for this purpose. The term 'good' refers to award of one of the top two UK grade bands (first or 2i class) for the undergraduate degree or when assessing a component (for more details see An exploration of the award gap in cell biology courses at University College London).

The terms 'course' and 'module' are used interchangeably and refer to a single body of work within a degree.

It is recognised that in some fields, measuring the results of a survey would be considered a quantitative method. However, for clarity, in this study a strict definition of quantitative evidence is used, where the raw source of information is itself numerical (for example, marks awarded). If the original source is answering questions, recounting lived or other experience, or providing opinions such as in a survey, interview, discussion or other type of feedback, this is defined as qualitative in this study.

### Students included and excluded from the study

UCL classifies a student as having completed a module if a non-zero mark is awarded for the module overall; and only these 'completed' students were included in the analysis. A small minority of included students (4.5%: 36/792) completed the course (non-zero mark awarded), but UCL did not return a mark for either the

exam (35 students), or exam and coursework (one student). These are almost certainly authorised extenuating circumstances, which permits exclusion of the component from the final mark awarded to the student or delays in updating the student record. UCL assesses the final mark for these students based on the components that they complete and any other relevant material.

Hence in the figures, total student number in the categories of final mark, exam and coursework may vary slightly (see exact N in *Figure 2—source data 1*). For completeness (detailed further in An exploration of the award gap in cell biology courses at University College London), N = 8 students (included) directly joined the year 1–2 pathway at year 2.

### Estimating number of students in the year 1–2 pathway that are in the year 3 study

From the year 2 data it was possible to determine how many students (112/250; 44.8%) in the year 1–2 pathway (cohorts 1–3) flowed into course A, B, or C in year 3. It was not possible to identify the year 3 destinations of the fourth cohort because these students had not yet completed their year 3 when records were extracted. The same proportion of 44.8% was used to approximate the number that would have flowed from the fourth cohort (44.8% of 94 year 2 students = 42). The exact N for each cohort is reported in *Figure 2—source data 1*.

From the year 2 data provided, it was not possible to track which students in the year 1–2 pathway took course D in year 3. However, using information on the degree programme that was included with the year 3 data, it was possible to identify a further 28 students that must have flowed from the year 1–2 pathway, resulting in a total of 182 (140 exact and 42 estimated) students that were in both studies.

### Student cohorts, student flows and interrupted studies

All students in this study are undergraduate students and they could select whether to graduate with a bachelor's (3 year course) or a master's (4 year course) undergraduate degree.

Here, >99% (N = 787/792) of students stayed with their cohort. For ease of quantification for the <1% of students who interrupted their studies, the data in the figures, tables, source data and supplements are parsed by the academic year in which each individual student took each individual course.

### Data extraction, research ethics and data protection

UCL department of Student Data extracted results from the university's Student Record, which includes the ethnicity of each student. The identity of each student is not available from the datasets supplied to the author. UCL legal services advise that in the context of external publishing, the data is considered fully anonymised and aggregated and thus is exempt from UK General Data Protection 2018 and the superceded Data Protection 1998 legislation and from Record of Processing Activity statutory requirement. Hence, they further advise explicit consent is not required. Students give their general consent through a privacy notice that they agree to when enrolling at UCL. The notice explains that their data, including special category (in this case ethnicity) and their marks awarded may be processed (such as this work here), and the legal basis for doing so. Here, that basis is to meet UCL's legal obligation of monitoring equality of opportunity and treatment (as is the case for all UK universities). Students may opt out of providing certain data such as gender and ethnicity and is recorded as 'unknown' by UCL.

UCL conditions to publish externally have been met: module code and precise titles are not provided, and undergraduate, PhD and postdoctoral data are anonymous and aggregated. The project is registered with UCL Data Protection and has UCL Research Ethics Committee approval (17663/001).

### Cell biology assessed components

The exams were invigilated and for the year 1 cell biology courses were mainly multiple-choice questions (machine-marked, student identity-blind) and for the year 2 and year 3 cell biology courses mainly long-answer essays (manually marked, student identity-blind). Students did not see the exam in advance. Coursework comprised one or more of the following types of activities: lab practical work, open-book work questions, discursive pieces, and/or a project (all marked blind).

### Quantification

In this study the following was calculated for the indicated cohort or course: the average A-level entrance tariff, proportion of each A-level grade awarded, the percentage of BAME and white students awarded a specified grade for different components of the course, and mean of all marks awarded to students on the course or in the cohort. The mean of all cohorts or courses in each year was then calculated along with the standard error of the mean (SEM). The difference in the mean percent of BAME and white students awarded a good mark (first or 2i) or first class mark-only was then used to determine the award gap for each year, except in the following case:

It was not possible to calculate the award gap in first class marks in this way for year two due to insufficient white and BAME students awarded first class marks in the exam component for each cohort. So instead the award gap was calculated based on the aggregated data of either three (cohorts 1, 2 and 3) or all four cohorts (noted in text).

Frequencies of raw marks awarded to BAME students were normalized to white students (*Figure 3*) by multiplying by factors 1.14–1.19 (provided in *Figure 3—figure supplement 1—source data 1*). This enabled fair comparison of these two ethnic groups where BAME student raw total population was smaller than white students.

### Assessing significance of results

Due to the relatively small number of cohorts or courses studied for each year (N = 4), a statistical test of significance between the different award gaps between white and BAME students were not calculated and the raw data were evaluated instead, as recommended (*Krzywinski and Altman, 2013*; *Vaux, 2012*).

### Attrition of BAME researchers occurs early in the academic trajectory to professor (Introduction)

This is an approximate guide as UK data, other than for undergraduates and full professors, are not precisely reported at the national level (discussed in *Policy 5b*). The academic trajectory was constructed from source data in section 3 of two reports published by AdvanceHE, which included all degree subjects and ethnicity of all UK academic staff and students (*AdvanceHE, 2020a* and *AdvanceHE, 2020b*).

Out of all the students and staff included in the report, the following percent self-identified as BAME: 23.69% of graduates (with a bachelor's or undergraduate master's degree); 18.07% of PhD students (proxy - PGR); 10.7% postdoc researchers (proxy – academic level K which includes researcher and research fellow, but also some other very junior academic staff); 10.9% of assistant professors (academic level J); 9.9% of

associate professors (academic level I); 9.1% of full professors (level '5A').

### Tracing the evidence underlying recommendations on how to close the award-gap (Policy 2)

The type of evidence that informed the 197 recommendations summarised in *Figure 4* was identified either from: *ECU and HEA, 2008*, *Berry and Loke, 2011*, *Amos and Doku, 2019*, or by back-tracing the evidence from other studies or reflective papers from participating higher education institutes, cited therein. These were carefully checked at the original source. Most recommendations seemed based on informal observation (74.62%; 147/197) and a few (4.57%; 9/197) could not be traced back to any obvious evidence. The other 41 recommendations were informed by data as described in the main text.

Ethnicity of respondents was not reported in 15/30 of surveys, interviews or discussions or other feedback included or cited in these three papers: *ECU and HEA, 2008*, *Berry and Loke, 2011*, *Amos and Doku, 2019*. Out of the 15 that did report ethnicity, 66.7% (10) only included responses from minority ethnic staff and students.

### Calculation of UK funding of award gap research (Policy 3)

UK Government Office for Students ('catalyst funding') and university (typically 'matched' funding) was identified from 'Addressing Barriers to Student Success Programme', which also includes funds that are specifically aimed at reducing the award gap (10/17 projects).

UCL was part of the Kingston University-lead consortium, which funded development and production of the 'UCL inclusive curriculum toolkit' and a small team to provide data and advise faculty deans. Total funding for the ten projects was £9,183,471 to be spent over typically three years (March 2017- October 2019). The British Medical Council has awarded an additional £85,000 to individual researchers. There are 360,650 UK-domiciled, BAME undergraduate students in the UK (Higher Education Statistics Agency). Thus, UK government and other funding is £8.49 per BAME student per year (and the funding has now ended).

### Calculation of funding per dementia patient (Policy 4)

The Dementia Discovery Fund is £250 million and there are about 850,000 dementia patients in the UK (as of 2021) giving £294 funding per patient.

### COVID-19 statement

This study predates the COVID-19 pandemic and all data (including all citations) are of original assessments and degree classifications as they stood prior to universities changing them in March 2020 as a result of the pandemic.

### Acknowledgements

Many thanks to Nadine Mogford, the Tetrad PhD programme office and Frank Bayliss for sharing data and details of their successful efforts to increase the proportion of minority ethnic students on the LIDo and UCSF PhD programmes, respectively. Thank you Lucy Newman for data on recruitment of PhD students to LMCB programmes. Data shared was either final impact (Nadine Mogford) or raw (all others) which were then quantified. Thanks to Kath Woolf for sharing background information in medicine, and Claire Herbert for the idea to refer to the 'attainment gap' as an 'award gap' (abstract, UK Education Conference 2019). Also, thanks to Ijeoma Uchegbu and Charmian Dawson for discussion on the UK award gap and Frank Bayliss for additional resources on the GRE; and Ijeoma Uchegbu, Rob de Bruin and Haley Flower for helpful comments on the manuscript.

**Louise Cramer** is in the MRC Laboratory for Molecular Cell Biology and affiliated with the Department of Cell and Developmental Biology, University College London, London, United Kingdom

l.cramer@ucl.ac.uk

https://orcid.org/0000-0003-3502-0434

*Author contributions:* Louise Cramer, Conceptualization, Formal analysis, Validation, Investigation, Visualization, Methodology, Writing - original draft, Writing - review and editing

*Competing interests:* The author declares that no competing interests exist.

*Ethics:* Human subjects: No explicit consent was required. This is described in detail in its own section in the methods.

## Funding

No external funding was received for this work. The UK Medical Research Council, grant code MCU12266B provided space to write parts of the manuscript.

**Decision letter and Author response**

Decision letter https://doi.org/10.7554/eLife.58971.sa1
Author response https://doi.org/10.7554/eLife.58971.sa2

## Additional files

### Supplementary files

• Transparent reporting form

### Data availability

The study looked at individual academic results of human subjects and for legal reasons can only be published as aggregated data. The individual grades of each student are therefore not included. Raw values used to generate the figures in the article have been included in the source data. See the methods section for further details.

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
