## [Decision Letter]

Thank you for submitting your article "Alternative strategies to reduce the race-gap in undergraduate degree scores and career progression" to *eLife* for consideration as a Feature Article.

Your article has been reviewed by three peer reviewers, and the evaluation has been overseen by two members of the *eLife* Features Team (Julia Deathridge and Peter Rodgers). The following individuals involved in review of your submission have agreed to reveal their identity: Tamsin Majerus (Reviewer #1); Kaushika Patel (Reviewer #2); Juliet Coates (Reviewer #3).

We have discussed the reviews and drafted this decision letter to help you prepare a revised submission.

Summary

This work provides a detailed analysis of the differences in grades awarded to white and Black, Asian or Minority Ethnic (BAME) students on a cell biology degree at a university in the UK. Unlike previous studies, this paper does not just look at the overall degree mark but at the individual assessments that make up the degree, and finds that exams contribute the most to the 'award-gap' in the degree studied. Although this finding may not be representative of all degrees, the methodology provides a framework that can be used to investigate the award-gap of other courses or programs, beyond the sciences, which may also have multiple exam and coursework components. The paper also provides general policy recommendations that will be of interest to leaders of Higher Education Institutes (HEIs) who are engaged in identifying and implementing strategies for closing the awarding gap. However, there are also a number of points that need to be addressed to make the article suitable for publication.

Essential revisions:

1) In the policy recommendations there is a strong emphasis on research funding and increased staffing. However, much research has already gone into 'why' the attainment-gap exists, and the recommendations should focus more on how HEIs can improve the experiences of BAME students. We, therefore, suggest removing policy 3, and adding the following recommendations to policy 2:

a) To spend more funding and resources on interrogating every process of the academy: the curriculum, appointment processes, assessments, the training delivered around racism (would probably fit into policy 2a).

b) To provide more training for academic staff so they are better at recognizing and calling out the toxic behaviours often experienced by BAME students, such as microaggressions. BAME students acutely feel a lack of similar role models in the academy and lack a sense of belonging. We, therefore, need to work on building supportive communities that really listen to the experiences of BAME students and collaborates with them on the process of making academia more inclusive so that barriers can be removed compassionately (Atkinson 2002).

c) Academic staff should also receive mandatory training in how to assess and grade work, avoiding implicit bias (techniques other than anonymising which is a good start, might include clear, specific rubrics and mark schemes). There is a reason that multiple-choice exams marked by machines tend not to show a bias and in part that is because the answer is either right or wrong, not subjective. This could be added to policy 2a.

2) For policy 4 to work, we need to find people who believe they will benefit from the change and put them in positions of power. Race is an uncomfortable subject for many who are unwilling (consciously or unconsciously) to confront their privilege. It might be worth adding a comment to that effect.

3) Please incorporate the following suggestions into policy 5:

a) For policy 5b to include removing the requirement for a 1st class degree to get a PhD place and to do shortlisting for recruitment based on anonymous CVs.

b) More resource needs to go into the training and diversity of recruitment panels.

c) Encouraging more BAME applicants will require more visible role models at higher levels but there is a worry around overburdening the minority, and this may be worth discussing.

4) Please link policy recommendations to pre-existing charter marks/professional awards. The two that are obvious are the Race Equality Charter (REC) and the Teaching Excellence Framework (TEF). A major step forward would be to require a REC award and clear evidence of action and plans to eliminate award gaps as fundamental criteria for the TEF.

5) Please remove the term 'race-gap' from the title and elsewhere in the text.

Using the term 'race-gap' is problematic for two reasons: first, it implies that there is a gap in race, not award (some places the text uses race award gap). The other issue is with the term 'race' itself. Mostly 'ethnicity' is used and this is a 'better' term. 'Race' has a problematic definition and its use simply muddies waters that need not be muddied! Please, therefore, stick to using 'ethnicity' and 'award-gap' consistently throughout the manuscript.

6) Please can you clarify the description of A level grades obtained by BAME and White students on the cell biology course (lines 233-236).

From Figure 1 A it seems as if the vast majority of students got one A* and 2As. Whereas the text (lines 233-236) suggests that students got a mix of A* and As. On the face of it then this could vary from 3A* grades to 3A grades (via A*A*A, A*AA). This leaves one question unanswered (except possibly by presumption that virtually all students regardless of ethnicity got A*AA) and that is was there a difference if the top grades are divided into the 5 possible combinations? It would help if this detail could be added/explained. This is important because the conclusion (lines 239-241) is only robust if this underlying assumption is correct.

7) In the literature review, please include more detail on how the awarding gaps for BAME students impact their 'academic' career progression. This will help strengthen the overall aim of the manuscript.

8) Please add the following to the Limitations section:

a) The methodology is clear and can be replicated but only where a combination of coursework and examinations make up the assessment components. If a degree, module, or specific subject within a degree only utilises one assessment type then the methodology will either not be relevant or will need adaptation.

b) One limitation of the study is perhaps that the cohorts, degree streams, and course selection is very complex as many different types of students take cell biology courses. Although the data is analysed appropriately using two different approaches, it means that any nuances around achievement for students on different degree programs (who may experience differently supportive learning environments) are masked.

c) The manuscript also aims to identify how the alternative strategy will look to reduce gaps in career progression. Although the outcomes of the study demonstrate the differences in awarding based on component assessments, it is not specific with regards to challenges for career progression for BAME graduates. The narratives around the impact of the awarding gap and the recommendations and policies do provide a direct link to how changing assessment methodologies could improve outcomes and in time career progression but this could be considered as a 'by-product' of the alternative strategy.

9) Another limitation is that the data analysis focuses on 'good honours', a first or 2.1 degree. We have carried out similar (unpublished) analyses at my institution that suggest that the 'first class' awarding gap is much bigger than that for good honours. Would it be possible to analyse firsts-only? The current analysis suggests that marking coursework identity-blind (largely) eliminates the awarding gap but (again based on my own data analyses) I wonder if looking at firsts-only would tell a different story.

If this is not possible, please include it as a limitation of the study in the revised manuscript and explain why you have decided not to address it in your point-by-point response to this letter.

10) To disaggregate the ethnicity data. "BAME" is a blanket term that includes groups in addition to people of colour. Moreover, previous data shows that Black students tend to fare worse than, for example, Asian students. Could you specifically look at the awarding gap for Black students?

If this is not possible, please include it as a limitation of the study in the revised manuscript and explain why you have decided not to address it in your point-by-point response to this letter.

Note: Here for your information are the original reviews, from which the above list of major comments have been taken.

*Reviewer #1:*

This is a very interesting and timely paper, discussing the important issue of the 'award gap' which is clearly an issue impacting on degree outcomes as well as career progression and opportunity. The paper provides welcome subject-level consideration and generalised policy recommendations to address the issues discussed. I absolutely think it should be published subject to the author addressing a small number of suggested corrections/amendments.*Reviewer #2:*

This manuscript will be of interest to all academics and leaders of Higher Education Institutes (HEIs) who are engaged in identifying and implementing a ranges of strategies for closing the awarding gap. The data presented in the manuscript clearly provides a basis for the full range of degrees who assess via both examination and coursework to consider the marking differentials between the two methods and their impact on either maintaining, worsening or closing the awarding gap.

The manuscript has the potential of significant impact across a range of degrees, beyond the sciences, due to its exploration of marks at component level and through the comparison of exams and coursework. It does provide a methodology which can be readily replicated across other disciplines and institutions.

A challenge to be considered is the emphasis within the recommendations (policies) around research funding and increased staffing. A great deal of insight has already been achieved with one of the key areas of focus landing on the levels of cultural competence and feelings of identity and belonging which need to be developed and improved within HEI's which although recognised within the recommendations does feature as less significant.

This manuscripts provides a clear overview of the 'race awarding' gap issue within western institutions and provides a clear outline of the range of research undertaken over a 10 year period with little progression in closing the race awarding gap.

The manuscript aims to identify alternative strategies in reducing the gap and certainly the methodology used, as outlined, does provide a different perspective in looking at assessment components. The methodology is clear and can be replicated but only where a combination of course work and examinations make up the assessment components. If a degree, module or specific subject within a degree only utilises one assessment type then the methodology will either not be relevant or will need adaptation.

The manuscript also aims to identify how the alternative strategy will look to reduce gaps in career progression. The outcomes of the study do demonstrate the differences in awarding based on component assessments but what the strategy does not do is be more specific with regards to challenges for career progression for BAME graduates. The narratives around the impact of the awarding gap and the recommendations and policies do provide a direct link to how changing assessment methodologies could improve outcomes and in time career progression but this could be considered as a 'by product' of the alternative strategy.

The HEI sector is currently becoming more aware of issues of racial inequality and the impetus generated by the Black Lives Matter movement both here and across the world has provided a spotlight on these racial inequalities in all sectors and particularly in higher education where the discrimination exacerbates overall life chances of a significant part of the university population. And so although the awarding/attainment gap has been researched, reviewed and action planned for many many years the BLM spotlight has provided a new energy and drive to address this significant inequality with a new and particular focus over the last few years away from the 'student deficit' model and more towards the 'institutional responsibility' model. So in this very current context this work will be particularly relevant in the wider 'race equality' context the sector finds itself in.

The manuscript would be strengthened to deliver on its aim if the connection to awarding gaps for BAME students and their 'academic' career progression were emphasised more in the literature review.*Reviewer #3:*

This paper provides a detailed breakdown of the awarding gap between BAME students and white students on a cell biology degree programme (with multiple pathways) at a single institution with a large BAME population. This detailed analysis shows that the biggest awarding gap lies in the exam component of the assessment and suggests some policies to close the gap going forward. There are some limitations to the analysis (should include first class marks as well as good honours) and suggested policies (e.g. vagueness around asking for funding) that I will address in detail in my review.In this study, the author examines the awarding gap that exists between BAME students and white students on cell biology degree programs. Unlike previous studies, the research does not just look at overall degree classification but examines individual assessments that make up the degree. The research concludes that it is specifically the exam component of assessment that contributes towards the awarding gap in Cell Biology degrees at UCL. Although these conclusions may not be representative of all programmes and institutions, the study provides a welcome framework to enable others to carry out this kind of detailed analysis if they have not started doing so already and shows that it is possible to carry out these analyses even in complex degree streams and pathways. From the data analysis, clear policy recommendations are put forward although I would caution against advocating more funding into research around 'why' the gap exists and instead repurpose such money for investing in proper resource to tackle the problem including by creating compassionate communities within institutions of BAME students working together with allies who have the power to instigate change.

Specific points – strengths, and issues to address, around data analysis.

1) The introduction sets the context for the pervasive awarding gap (and other factors that may contribute to it) extremely well and raises the issue of variability between subjects.

2) The data is collected from seven student cohorts across all three years of the degrees, including only students who completed the course. All marking for this dataset was carried out by machine or candidate identity-blind, which is a strength of the institution's assessment process as it eliminates human bias, conscious or unconscious.

3) One limitation of the study is perhaps that the cohorts, degree streams, and course selection is very complex as many different types of students take cell biology courses. Given that, the data is analysed appropriately using two different approaches, but that does mean that any nuances around achievement for students on different degree programs (who may experience differently supportive learning environments) is masked.

4) Another limitation is that the data analysis focuses on 'good honours', a first or 2.1 degree. We have carried out similar (unpublished) analyses at my institution that suggest that the 'first class' awarding gap is much bigger than that for good honours. Would it be possible to analyse firsts-only? The current analysis suggests that marking coursework identity-blind (largely) eliminates the awarding gap but (again based on my own data analyses) I wonder if looking at firsts-only would tell a different story. I think that this would be an important addition to the study.

5) I wonder if it is also possible (as this is a large dataset) to disaggregate the ethnicity data. "BAME" is a blanket term that includes groups in addition to people of colour. Moreover, previous data shows that Black students tend to fare worse than, for example, Asian students. Could we specifically look at the awarding gap for Black students is the largest?

6) A thought about the statistical analysis: a t-test is not appropriate for non-parametric (%) data unless normality has been tested for or a correction has been applied. Has this been done? It is not clear from the methods.

7) Showing figure 2c as continuous data points joined by a line is unclear: it would be more consistent to use bars as in Figure

Some thoughts about the discussion and policy recommendations.

I absolutely agree with the suggestions of policy 1 and policy 2. However, Policy 3 needs to be careful and specific otherwise it is not convincing to "Increase Direct Funding to Research the Factors that Cause the Award Gap", in fact I would be tempted to remove this section and expand Policy 2 as outlined in the next couple of paragraphs.

Much "why" research has already been carried out over the last 30 years that all points in the same direction: there are huge systemic, structural barriers that are put in place by people of privilege and power that disadvantage BAME students in education and the problems start before University, although are reinforced in higher education. It is the 'business as usual' approach has to change as this is what causes the problem (see, for example, much of David Gillborn's research including Gillborn et al. (2017)). Racism is subtle, common and can be well intentioned/paternalistic ('we know best how to fix you'). Funding and resource would be better spent on Policy 2 and in addition an interrogation of every process of the academy: the curriculum, appointment processes, assessments, the training delivered around racism – this would probably fit under Policy 2a.

More thoughts on Policy 2a: We need to resource local support that involves the relevant students. Building supportive communities is key. We need to really listen to student experiences of all the academy's processes and involve students in the process of change as collaborators so that barriers can be removed compassionately (Atkinson 2002). My own anecdotal listening plus institutional training shows that BAME students are subject to daily micro-aggressions and more training is needed to increase the number of allies who are confident to call out toxic behaviour. BAME students acutely feel a lack of similar role models in the academy and lack a sense of belonging. Thus, changing recruitment processes is a key step (see policy 5 below).

For policy 4 to work, we need to find people who believe they will benefit from the change and put them in positions of power. Race is an uncomfortable subject for many who are unwilling (consciously or unconsciously) to confront their privilege. It might be worth adding a comment to that effect.

Policy 5: I absolutely agree. I would advocate for policy 5b to include removing the requirement for a 1st class degree to get a PhD place (this is partly why I made my earlier point about extending the data analysis beyond good honours if at all possible) and to do shortlisting for recruitment based on anonymous CVs. More resource needs to go into the training and diversity of recruitment panels. Encouraging more BAME applicants will require more visible role models at higher levels but there is a worry around over burdening the minority with this work perhaps this could be discussed a little too.

References:

Atkinson J (2002) Trauma Trails, recreating song lines: The transgenerational effects of trauma in indigenous Australia p.13-17 (Spinifex Press).

Gillborn D et al. (2017) Moving the goalposts: Education policy and 25 years of the Black/White achievement gap. British Educational Research Journal 43 p.848-874.

---

## [Author Response]

[Editor’s note: Further changes were made following revisions to the manuscript in response to queries and discussions with the editor. As a result, some of the comments below may differ from the published version of the article.]

Essential revisions:1) In the policy recommendations there is a strong emphasis on research funding and increased staffing. However, much research has already gone into 'why' the attainment-gap exists, and the recommendations should focus more on how HEIs can improve the experiences of BAME students. We, therefore, suggest removing policy 3, and adding the following recommendations to policy 2:a) To spend more funding and resources on interrogating every process of the academy: the curriculum, appointment processes, assessments, the training delivered around racism (would probably fit into policy 2a).b) To provide more training for academic staff so they are better at recognizing and calling out the toxic behaviours often experienced by BAME students, such as microaggressions. BAME students acutely feel a lack of similar role models in the academy and lack a sense of belonging. We, therefore, need to work on building supportive communities that really listen to the experiences of BAME students and collaborates with them on the process of making academia more inclusive so that barriers can be removed compassionately (Atkinson 2002).c) Academic staff should also receive mandatory training in how to assess and grade work, avoiding implicit bias (techniques other than anonymising which is a good start, might include clear, specific rubrics and mark schemes). There is a reason that multiple-choice exams marked by machines tend not to show a bias and in part that is because the answer is either right or wrong, not subjective. This could be added to policy 2a.

As suggested these reviewer recommendations around funding (1a), resources (1a) and training (1a-c) have been incorporated into extended Policy 1 (old Policy 2) and elsewhere: addition to Policy 4; new Figure 4; within the main conclusion. In the original manuscript previous recommendations on how to close the award-gap were restricted as a whole to citations of previous studies to enable the current manuscript to focus on new ideas. However, the point is well taken that may have inadvertently given less space to a key anchor “Widespread or sustained delivery of the reported solutions can’t be working sufficiently then…” which is now addressed.

Of note, a critical objective in the revised manuscript was to fully meet these reviewer suggestions on previous recommendations, but also to ensure the manuscript simultaneously remained focused on addressing “why the award-gap is only closing slowly”.

This aim seems met by providing a summary of all previous recommendations distributed by category (new Figure 4). This was achieved by careful assessment and quantification of three, major substantive reports representing the past decade (nearly 200 recommendations). In addition, it was thought important to capture substantive previous work in a balanced way (to represent across all ideas as well as the reviewer ones suggested) and this is reflected in Figure 4.

New Figure 4 explicitly includes the category suggested by the reviewer (comment 1, improve the experiences of BAME students).

In new Figure 4, legend non-exhaustive examples of the top three categories of previous recommendations (infrastructure, student experience, teaching and learning) are listed. The reviewer suggestions to provide resources and training (reviewer comments 1a-c), are included as important examples within this list, and specifically all the ones listed by the reviewer in 1a-c.

The reviewer comment, “Spend more funding…” (start of comment 1) is taken to mean invest in existing recommendations to ensure resources and training (comment 1a-c) is implemented, is now an explicit part of the revised version: raised as an issue in the introduction to the policy section; in the title of Policy 1; in the introduction to Policy 1; in the content of new Policy 1d; and in the title and content of modified Policy 4.

Responding to the other part of reviewer comment 1 **“…**much research has already gone into 'why' the attainment-gap exists…” and “We, therefore, suggest removing policy 3,..”.

This comment is understood; there has been a lot of research and these were cited for example in the original manuscript and now stated more directly in the revised version “At first glance it may appear there is plenty of evidence on causes of the award-gap… However, on closer inspection there seem two large gaps in the evidence base.”

In retrospect perhaps, more details on how these evidence-gaps were originally identified and more clarity on the concept of ‘unexplained’ ethnic award-gap would have been helpful and both these are now expressed more clearly in the revised version.

Unexplained ethnic degree award-gap: Previous research (cited in the introduction) tested >20 different external factors (for example, poverty, age, study habits, type of previous school) as possible explanations for why the ethnic degree award-gap exists, but found that a large gap remained; this larger gap is known as the unexplained award-gap. It is thought to arise through being at university and is the one that universities have been charged to remove and the focus of the manuscript. This is now made more explicit in the introduction (literature review) of the revised version and in the introduction to the policy section.

Gaps in the evidence base: it is now made explicit, that gaps in the evidence are around those that seek causes of the unexplained (university-driven) award-gap: in the introduction to the policy section; in the title of new Policy 2 (old Policy 3); in the titles of new Figures 4 and 5. These identified gaps in the evidence are limited diverse types of evidence (in particular quantitative as defined by the raw data is numerical) and limited scientific approaches as defined by the principle of conducting a ‘fair test’; both these are reflected in the new title to Policy 2.

How limited diverse types of evidence and limited scientific approaches were identified:

i. Limited quantitative approaches were identified by careful and thorough mapping the types of evidence that underlies ~200 recommendations, representative of the past decade (new Figure 5), which have previously suggested to universities how to reduce the unexplained award-gap (new Figure 4).

Please note, it is acknowledged in the revised manuscript that “Clearly, much has been quantified, particularly the size of award-gaps, however to date this has almost exclusively been used as evidence that the problem exists, as a benchmark or as attempts to identify the size (if any) of a number of potential, contributory external factors […] Yet, these do not in themselves identify the larger, university-driven (unexplained) causes of the gap.”

Note it is also acknowledged in the revision, that whilst this investigation has identified that quantitative approaches seem under-represented in seeking causes of the unexplained gap (it was found to be only ~0.5% of the evidence base), it is also made plain in the text to Policy 2, that the existing main evidence (qualitative: as defined by the raw data is dialogue with students and staff) are critical too. The point is that to close the unexplained gap faster (one of the main aims of the paper) likely requires a balance in quantitative and qualitative approaches.

ii. When this research was started, there did not seem many other scientific approaches in the literature. This was also recently observed by a large collaboration between the National Union of Students, Universities UK and 99 HEIs. Baroness Amos and Amatey Doku conclude “Universities need to take a far more scientific approach to tackling the attainment gap…” as one of the top 5 outstanding actions required to successfully reduce the award-gap (page 2 in Amos and Doku 2019).

There are several benefits to increasing the diversity of approaches and scientific evidence: they have the potential of contributing to reducing the award-gap in ways less amenable by other methods; one example is the findings in this paper (Figure 1-3) another is facilitating impact-evidence on what works, and equally important can guide prioritisation of allocation of valuable resources (for further details, please see Policy 2).

This manuscript might contribute to awareness that these gaps in the evidence base seem to exist; but practically, funding is required to address it. This is highlighted in modified original Policy 3 and it is then suggested that creation of a funding alliance could generate the necessary funds in Policy 4.

Note, it is also suggested in the revised version that this alliance could also be used to fund the implementation of existing recommendations and resources (as raised by the reviewer in comment 1) and scientifically test which ones work.

2) For policy 4 to work, we need to find people who believe they will benefit from the change and put them in positions of power. Race is an uncomfortable subject for many who are unwilling (consciously or unconsciously) to confront their privilege. It might be worth adding a comment to that effect.

Absolutely. In the original manuscript it was proposed that the Nuffield Foundation house such an alliance, exactly for this reason. The revised text now makes plain this point, “Strong leaders who believe what society has to gain from such an alliance are required for success; it is proposed then to take advantage of a pre-existing structure such as the UK’s Nuffield Foundation, which already seems to demonstrate such a belief both in its strategy document 2017-2021 and it its funding of projects widely across education, welfare and justice; their housing of the suggested alliance may speed up its creation.”

3) Please incorporate the following suggestions into policy 5:a) For policy 5b to include removing the requirement for a 1st class degree to get a PhD place.

This was in the original text, it is now rephrased to make clearer: “…have removed the highest undergraduate degree class (1^st^ class) as advantage for selection for interview or award of a PhD place (LIDo)”.

And to do shortlisting for recruitment based on anonymous CVs.

This is an important point and has been attempted by some PhD programmes. To address the reviewer’s point, a sentence has been added to policy 5b: “It is also suggested that candidates’ names are removed from CVs and LIDo have had partial success with anonymous CVs; the main issue remaining is how to ensure 100% of referees use applicant number instead of name in their letters of recommendation.”

b) More resource needs to go into the training and diversity of recruitment panels.

This is now added to a new section in policy 5b:

“(iv) In part, the success of these three programmes is that there seems specific training of recruitment panels and at least for some, a common rubic for selection of applicants. Diversity on selection panels is an ongoing concern – it directly relates to the high loss of potential senior academics early in the pipeline to professor (detailed in the introduction) and some universities (including UCL) are implementing initiatives to increase representation of minority ethnic staff on panels.”

c) Encouraging more BAME applicants will require more visible role models at higher levels but there is a worry around overburdening the minority, and this may be worth discussing.

Role models seem inherently part of the success of policy 5bii (establishing relationships between highly competitive PhD programmes and HEIs where minority ethnic undergraduate students are more likely to attend) and appear to work without any sense of overburden. At UCSF, minority (and other) ethnic doctorate students have the opportunity to visit minority serving colleges, some of which were their own undergraduate colleges to reinforce the cycle of success in informal ambassador type roles. At Bridge, there is a cycle of visible mentorship which seems to function in a similar way. At LIDo PhD students serve as ambassadors at interview days and mentor undergraduate students on summer placement schemes, whom are recruited mainly from lower tariff universities. Space limitations probably preclude these details, however the point is noted and encompassed in additional text: “An added benefit of these schemes is that they incorporate opportunities for current minority ethnic doctorate students on the PhD programmes to act as role models in such a way that also contributes to their own career development.”

4) Please link policy recommendations to pre-existing charter marks/professional awards. The two that are obvious are the Race Equality Charter (REC) and the Teaching Excellence Framework (TEF). A major step forward would be to require a REC award and clear evidence of action and plans to eliminate award gaps as fundamental criteria for the TEF.

There is some hesitation here from the experience of working on the Athena Swan (AS) charter marks (our department was one of the first in the UK to receive Gold in 2016 and was awarded Gold again in 2020). What seems to have been discussed at AdvanceHE (who run the AS and RE charter marks) is the acknowledgement of substantial and disproportionate burden on female staff engaging in the AS charter mark work (which is very high). Without explicitly linking contributions to these charter marks to promotion criteria, may result in similar disproportionate overburden for minority ethnic staff working on the REC/TEF. Some (if not all) HEIs include an expectation of citizenship in promotion applications, however there seems variability – both to the weighting/clarity of how much that work counts to promotion and that some HEIs do not appear to permit promotion on contribution to citizenship alone.

Another possible idea is to link the REC as a requirement to grant applications. However, that too has now changed; whereas a silver AS award was required, some participating grant award bodies have now (reported locally in 2020) withdrawn the requirement, no reason was given; possibly connected to disproportionate load or other reasons.

The reviewer suggestion could work very well (gains in reducing the gender professor-gap may have come from similar links in the AS charter marks), however space would be required in the manuscript to discuss these other significant factors.

(Somewhat related, however, the REC and fellow of HEA awards are now linked to a point in Policy 2).

5) Please remove the term 'race-gap' from the title and elsewhere in the text.Using the term 'race-gap' is problematic for two reasons: first, it implies that there is a gap in race, not award (some places the text uses race award gap). The other issue is with the term 'race' itself. Mostly 'ethnicity' is used and this is a 'better' term. 'Race' has a problematic definition and its use simply muddies waters that need not be muddied! Please, therefore, stick to using 'ethnicity' and 'award-gap' consistently throughout the manuscript.

Ethnic is used throughout the manuscript, other than where it is a title such as Race Equality Charter.

6) Please can you clarify the description of A level grades obtained by BAME and White students on the cell biology course (lines 233-236).From Figure 1 A it seems as if the vast majority of students got one A* and 2As.

Indeed, the average tariff was the equivalent of A*AA for almost all students. This is now directly shown in new Figure 1A. To be able to determine the average entrance score for each individual student, grades were converted to the UK standard tariff of: A* = 56; A = 48; B = 40 (no student obtained lower than a B), then the cohort mean determined:

“The mean admission tariff for these BAME (50.53+/- 0.15) and White (50.29+/- 0.33) students were virtually identical (Figure 1A): the equivalent of nearly all students in each of these ethnic groupings gaining one A* grade and two A grades (which would be a mean tariff of 50.7). These data are very similar to the mean entrance tariff for all BAME (50.27 +/-0.06) and all white (50.03 +/- 0.01) students at UCL who enter with A levels for the same cohort years as these cell biology students (2013, 2014, 2015 entrants), thus in this context cell biology students in this study are typical for the university.”

Whereas the text (lines 233-236) suggests that students got a mix of A* and As. On the face of it then this could vary from 3A* grades to 3A grades (via A*A*A, A*AA).

It is realised that the description here could have been confusing. What was quantified is the total number of each grade to each cohort (in other words what was the total number of A grades awarded to all the BAME students entering in 2013, then in 2014, 2015 and repeated for all the different grades). The text is now rephrased and the data is now in Figure 1B, as:

“An alternative approach to assess admission qualifications is to look at the distribution of the total number of each individual grade (A*, A, B, C or lower) awarded across each cohort: as expected distributions for BAME and White students were nearly identical (Figure 1B); of the total grades awarded to BAME students 95.45% +/- 1.01 were A* or A, and to White students this figure was 95.02% +/- 0.61 (Figure 1C).”

This leaves one question unanswered (except possibly by presumption that virtually all students regardless of ethnicity got A*AA) and that is was there a difference if the top grades are divided into the 5 possible combinations? It would help if this detail could be added/explained. This is important because the conclusion (lines 239-241) is only robust if this underlying assumption is correct.

A*AA was the equivalent tariff for virtually all students; please see new data in new Figure 1A, described above.

7) In the literature review, please include more detail on how the awarding gaps for BAME students impact their 'academic' career progression. This will help strengthen the overall aim of the manuscript.

More details have been provided, in particular: how loss of potential minority ethnic professors is directly linked to less competitive undergraduate degree scores, is more clearly explained; and more details on how selection criteria for PhD positions (which create barriers for minority ethnic students) is directly linked to data and suggested actions in the paper. Other changes to the title, abstract and introduction to the policy section also help strengthen the overall aim of the manuscript in this context.

8) Please add the following to the Limitations section:a) The methodology is clear and can be replicated but only where a combination of coursework and examinations make up the assessment components. If a degree, module, or specific subject within a degree only utilises one assessment type then the methodology will either not be relevant or will need adaptation.

The limitations section has been modified to incorporate the reviewer comment:

“The study is restricted to certain courses at one university. The wider applicability of the written exam and coursework findings, require testing in cell biology in other universities and in other subjects across institutions. Most degree subjects (if not all) are composed of different types of activities and assessed components, the method described here is also envisaged to be readily adaptable to look systematically at whichever ones are relevant to any particular subject.”

b) One limitation of the study is perhaps that the cohorts, degree streams, and course selection is very complex as many different types of students take cell biology courses. Although the data is analysed appropriately using two different approaches, it means that any nuances around achievement for students on different degree programs (who may experience differently supportive learning environments) are masked.

This good point seems more appropriate for the Discussion section as relative award-gap sizes are not limited when comparing the exam and coursework components to each other; the same student on their particular degree pathway would be in the same learning environment for both components. However different supportive environments may explain relative gap-sizes when comparing the same component-type between year-1, -2 and year-3 cell biology courses. This possibility has been added to other possibilities in the discussion that might explain relative gap-size in this context:

“It is also not yet possible to distinguish whether precise size of the exam and coursework award-gaps comparing the same component across the different year-1, -2 and year-3 cell biology courses (observable in Figure 2E, compare years; and perhaps also Figure 2D) is a manifestation of a difference in the precise curriculum activity on that course or how it is delivered or for example, other factors such as supportive environments that students may encounter differently on different degree routes or several of these factors combined.”

c) The manuscript also aims to identify how the alternative strategy will look to reduce gaps in career progression. Although the outcomes of the study demonstrate the differences in awarding based on component assessments, it is not specific with regards to challenges for career progression for BAME graduates. The narratives around the impact of the awarding gap and the recommendations and policies do provide a direct link to how changing assessment methodologies could improve outcomes and in time career progression but this could be considered as a 'by-product' of the alternative strategy.

This has been addressed by clarifying the aims of the paper in the title, abstract, introduction and policy section and more clearly describing direct links between the award-gap and academic progression (as explained in response to comment 7). In brief, one of the main purposes of the paper is to provide an account of ‘what works’ to increase academic career progression and to convey the concept of the need for parallel action. In other words, careers matter now; hence it is suggested to i) address the award-gap by asking what could speed progress which, in time would then directly contribute to career progression, as noted by the reviewer; and ii) whilst the award-gap still exists also take parallel (separate) action on addressing the career-gap directly. In the USA funding schemes to promote academic career progression for example seem more widely developed and discussed (cited in the paper).

Please find an extract from the revised introduction, as an example of how these comments have been addressed: “Then by focusing on asking the question of why the UK ethnic award-gap is only shrinking very slowly, some likely causes of the slow pace are identified. […] Given this award-gap directly reduces chance of career progression, dialogue with PhD programme organisers or analysis of published initiatives that have successfully increased proportion and retention of under-represented, minority ethnic doctorial students are also presented.”

9) Another limitation is that the data analysis focuses on 'good honours', a first or 2.1 degree. We have carried out similar (unpublished) analyses at my institution that suggest that the 'first class' awarding gap is much bigger than that for good honours. Would it be possible to analyse firsts-only? The current analysis suggests that marking coursework identity-blind (largely) eliminates the awarding gap but (again based on my own data analyses) I wonder if looking at firsts-only would tell a different story.If this is not possible, please include it as a limitation of the study in the revised manuscript and explain why you have decided not to address it in your point-by-point response to this letter.

This has been addressed with a substantial new body of work in the year 2 and year 3 cell biology courses (Figure 3 and Figure 3—figure supplement 1). The possibility of potentially hidden larger, coursework award-gaps was tested in the original manuscript, however a considerably deeper and wider test going beyond 1^st^ class grades, seeking any hidden gaps in the entire range of marks awarded from <30% to 100% in 5% intervals was performed and no larger gap in the coursework could be found (Figure 3, compare histograms).

As expected from these frequency distributions, the coursework award-gap remains small in either the analysis of 1^st^ class marks-only or all good marks combined and at least 2-fold smaller than the exam award-gap (which is also smaller at 1^st^ class marks-only) and the coursework award-gap at 1^st^ class-marks is not significant (Figure 3—figure supplement 1; Figure 3—figure supplement 1-Source data 1).

It was not feasible to fully assess year 1-cell biology though there was also no significant gap in the coursework in 1^st^ class marks in the available data and this is explained in the limitation of the study.

Overall, these new analyses and the original ones strengthen the view that the award-gap in these 6 cell biology courses is largely derived from the exam.

It was not possible to compare these new data to the unpublished work the reviewer mentioned, however, the UK average is that the award-gap at the level of 1^st^ class marks-only is *smaller* than at all good marks (AdvanceHE 2019b) which is consistent with the findings in these cell biology courses (above) (Figure 3 and Figure 3—figure supplement 1). None-the-less an assessment of other UK statistics (also in AdvanceHE 2019b) and other data at UCL, both looking at a finer grain reveals that there are at least 5 different patterns evident on the relative contribution from 1^st^ class marks and 2i marks to the overall ethnic award-gap in good grades – and one of these other patterns might be consistent with the unpublished data mentioned by the reviewer. This might be expected when looking at a finer grain (and this point and the existence of these 5 other patterns is acknowledged in the new version of the manuscript).

10) To disaggregate the ethnicity data. "BAME" is a blanket term that includes groups in addition to people of colour. Moreover, previous data shows that Black students tend to fare worse than, for example, Asian students. Could you specifically look at the awarding gap for Black students?If this is not possible, please include it as a limitation of the study in the revised manuscript and explain why you have decided not to address it in your point-by-point response to this letter.

3% of the population has a Black ethnic background (2011 Census) similar to the ~2% of the cell biology students in this study. The study-size would therefore have to be far larger to be able to assess Black ethnic students-only. Some preliminary findings are reported in the limitation section of the paper:

“Preliminary data, which is consistent with the UK pattern, looking at year-3 cell biology courses, the proportion of Black students awarded good grades for the final course mark (~38%) and for the exam (~38%) was up to 2-times lower than all minority ethnic students (BAME) for these two components (72.7% +/- 3.7 and 64.1% +/- 2.6 respectively). However, the proportion of Black students awarded good grades for cell biology coursework (~87%) was at least equally as high as all BAME students (82.1% +/- 2.0) and White students (87.5% +/- 1.5).”